# Scalable nano-architecture for stable near-blackbody solar absorption at high temperatures

Yifan Guo[1,2], Kaoru Tsuda[3], Sahar Hosseini[1,2], Yasushi Murakami[4], Antonio Tricoli[5,6], Joe Coventry[2], Wojciech Lipiński[7] & Juan F. Torres [1]✉

Light trapping enhancement by nanostructures is ubiquitous in engineering applications, for example, in improving highly-efficient concentrating solar thermal (CST) technologies. However, most nano-engineered coatings and metasurfaces are not scalable to large surfaces (> 100 m²) and are unstable at elevated temperatures (> 850 °C), hindering their wide-spread adoption in CST. Here, we propose a scalable layer nano-architecture that can significantly enhance the solar absorption of an arbitrary material. Our electromagnetics modelling predicts that the absorptance of cutting-edge light-absorbers can be further enhanced by more than 70%, i.e. relative improvement towards blackbody absorption from a baseline value without the nano-architecture. Experimentally, the nano-architecture yields a solar absorber that is 35% optically closer to a blackbody, even after long-term (1000 h) high-temperature (900 °C) ageing in air. A stable solar absorptance of more than 97.88 ± 0.14% is achieved, to the best of our knowledge, the highest so far reported for these extreme ageing conditions. The scalability of the layer nano-architecture is further demonstrated with a drone-assisted deposition, paving the way towards a simple yet significant solar absorptance boosting and maintenance method for existing and newly developed CST absorbing materials.

Concentrating solar thermal (CST) technologies are appealing renewable energy sources due to their inexpensive solar thermal energy storage and potential in direct high-temperature heating for a wide range of energy-intensive industrial thermal processes such as iron making[1,2]. For example, 16.8% of the entire global energy consumption is for high-temperature industrial processes[3]. Without addressing industrial thermal emissions, global warming cannot be kept within 1.5 °C[4]. Therefore, CST technologies are expected to have a key role in a sustainable future. Solar receivers are essential in CST technologies as they convert concentrated solar irradiation into heat. Receivers must be coated with efficient sunlight-absorbing materials to enhance solar–thermal energy conversion while operating at high temperatures. These solar absorber coatings need to be durable and have outstanding light-trapping properties[5]. Light-trapping enhancement has received extensive interest in many disciplines, especially in solar energy technologies. Among the methods for boosting sunlight absorption, nanostructures such as plasmons[6] have been widely reported. Nanostructures are promising due to a large solar absorption enhancement brought by light–matter interactions[7,8] with minimal damage to the underlying material[9].

[1]ANU HEAT Lab, School of Engineering, Australian National University, Canberra, Australia. [2]Thermal Energy Group, School of Engineering, Australian National University, Canberra, Australia. [3]Nano Frontier Technology, Tokyo, Japan. [4]Faculty of Textile Science and Technology, Shinshu University, Ueda, Japan. [5]Nanotechnology Research Laboratory, Faculty of Engineering, University of Sydney, Sydney, Australia. [6]Nanotechnology Research Laboratory, Research School of Chemistry, Australian National University, Canberra, Australia. [7]The Cyprus Institute, Nicosia, Cyprus. ✉e-mail: felipe.torres@anu.edu.au

Nanostructures have been widely investigated as means of improving CST coatings[10–17]. Current CST coatings have nanostructures that are generally produced by a curing process in which organic material is burnt, such as engineered polymer beads[18] or a binder as in Pyromark 2500®[19] (or "Pyromark"); the latter coating is considered to be the gold standard in CST. These nanostructured coatings are scalable to large surfaces, yet their optical characteristics are difficult to control. Therefore, attention has been put on sophisticated deposition methods to accurately tune the nanoscale architecture (or "nano-architecture"), as demonstrated in solar cells[9,20,21] and metamaterials such as electromagnetic absorbers[22,23]. Numerical and experimental results for nanostructure control in CST applications have been reported[24,25]. However, these deposition methods generally rely on rather complex fabrication processes such as lithography that require vacuum chambers limited in dimension to around 3 m[24] and thus are not scalable to conventional receivers, which are generally larger than 10 m[26]. In addition, most nanostructures are unstable at elevated temperatures in air[12,13] due to phase re-arrangement and transition[19], alteration of material composition[16] and sintering or crystal growth[27]. Although solar absorptance, a standard metric to evaluate the sunlight absorption[28,29], remains rather high for some conditions of long-term high-temperature ($\geq 600$ °C) isothermal ageing of Pyromark coatings, e.g. 96.2% after ageing for 3000 h at 800 °C[19] and 94.6% after ageing for 2350 h at 850 °C[10], the nanostructure morphology (and consequently the solar absorptance) is significantly affected at temperatures exceeding 850 °C[27]. This is because most nanomaterials, which are extended throughout the entire light-absorbing volume, tend to sinter at temperatures greater than 850 °C, severely affecting their durability and degrading the initially high solar absorptance[16].

Spherical photonic crystals have been evaluated and shown to be promising in numerous applications outside CST[30]. For low-temperature implementations, material for spherical nanoparticles (or "nanospheres") has been varied from a metal[31,32] based on plasmonic interaction[33] to dielectric[34,35]. To achieve an enhancement in sunlight absorption by light–nanosphere interaction, nanospheres have been placed either on the surface or within the material, as investigated in photovoltaic cells[36], but these nanomaterials are generally not stable in air at high temperatures. In our recent work[37] focused on high-temperature CST, we applied a controllable and scalable layer nano-architecture—or "nanolayer"—made of silica (i.e. a highly-stable ceramic at elevated temperatures) onto several arbitrary high-temperature solar absorber materials to permanently enhance their solar absorption in a broad wavelength spectrum, aiming at approaching the theoretical blackbody absorption value of 100%. However, the mechanisms associated with the absorptance improvement are not well understood, thus a pathway towards optimisation has been elusive. To enhance the solar absorptance at a minimal cost, i.e. without developing entirely new solar absorber materials and structures, more theoretical insight into light–nanolayer interaction is needed.

Furthermore, accelerated ageing tests have revealed that absorber materials generally degrade rapidly in a high-temperature environment in the presence of air due to the severe oxidisation of the underlying metallic substrate[13,27]. To ameliorate the degraded optical properties, a maintenance procedure must be devised and re-painting is generally conducted[38]. However, re-painting large solar receivers is not only challenging and expensive, but also increases coating thickness leading to excessive thermal barriers[16]. Thermal barrier refers to the temperature drop across the coating, which increases with coating thickness or decreases with thermal conductivity. Large thermal barriers are a serious problem in CST technologies because a large temperature drop through the coating may result in an excessive radiative heat loss[16] when achieving the target temperature in the heat transfer fluid (which is regulated by controlling its flow rate). Therefore, a

maintenance method that does not require the re-coating of the entire receiver and has a marginal effect on the thermal barrier is needed. In addition, the durability of such maintenance material should be experimentally assessed with long-term high-temperature ageing testing.

Here, we propose and investigate an approach to introducing nanoporous textures via a layer nano-architecture with highly controllable features, as shown in Fig. 1a–c, which is different to having an extended nanoporosity throughout the solar absorber. Compared to the thickness of the underlying absorber material—normally greater than 10 μm—the nanolayer is ultrathin, having a thickness normally smaller than 250 nm, i.e. less than 2.5% of the absorber material thickness. The nanolayer developed here is comprised of nanoparticles and a matrix that binds them onto the underlying material, which could be a solar absorber coating or an arbitrary substrate. Our proposed nano-architecture facilitates light absorption control because the nanosphere diameter and matrix thickness serve as simple 'dials' to tune light–matter interaction. In this study, we conduct a theoretical investigation into the optical performance of our proposed layer nano-architecture on different underlying solar absorber materials when varying design parameters such as nanosphere size. Then, we fabricate and test the nanolayer performance to obtain experimental evidence that the nano-architecture is beneficial in CST, both in pristine condition and after long-term thermal ageing. Finally, we demonstrate the scalability of our nanolayer by conducting a spray deposition using a drone or unmanned aerial vehicle (UAV) to improve the optical performance of existing high-temperature CST coatings. The scalability is key for the wide adoption of the proposed nano-architecture in the CST field. We show that our proposed layer nano-architecture can improve sunlight absorptance while being durable, which has the potential of serving as an upgrading, refurbishment or maintenance method for existing or degraded solar absorbers.

## Results
### Nanolayer effectiveness: a metric for quantifying proximity to blackbody absorption

Experimental results in Fig. 1d (visible range indicated by the rainbow band) show that a nanolayer, comprised of monodispersed silica nanospheres and a silica matrix, improves the spectral absorptance when deposited on three significantly different light absorbers[37]: a coral-structured coating[16] (with one of the highest reported solar absorptance), Pyromark coating[19] and Inconel 625 substrate. The coral-structured coating[16] is a stony-coral-inspired, hierarchical, high-temperature solar absorber with a large hierarchical range containing macro-scale protrusions of ca. 80 μm and micropores of ca. 3 μm (see Methods for more details on underlying solar absorber coatings). Light trapping is enhanced via multiple light reflections within these coral-structured features. After depositing the nanolayer (Fig. 1d), the absorptance is enhanced almost for the entire wavelength range between 250 nm and 2500 nm where most of the sunlight energy exists (ca. 99%[39]), for the three underlying absorbers.

To measure the improvement by the layer nano-architecture, we define a metric termed "nanolayer effectiveness" $e_{nl}$. This metric is used to characterise the reduction in sunlight reflection (i.e. increase in sunlight absorption) from the baseline value without nanolayer to the ideal case of a blackbody absorber for which there is zero light reflection:

$$e_{nl} = \left(1 - \frac{\rho_{nl}}{\rho_{sm}}\right) \times 100\%, \qquad (1)$$

where $\rho_{nl}$ is the solar reflectance of the sunlight-absorbing material with nanolayer on its surface and $\rho_{sm}$ is the solar reflectance of the underlying solar material without nanolayer (see Methods for definition of solar reflectance). The latter serves as a baseline value.

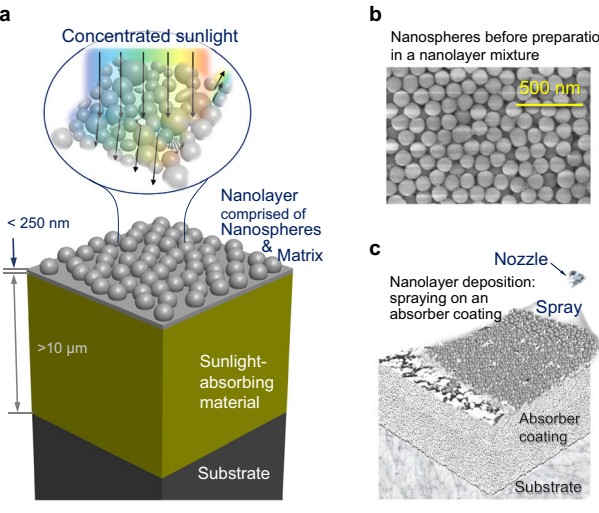

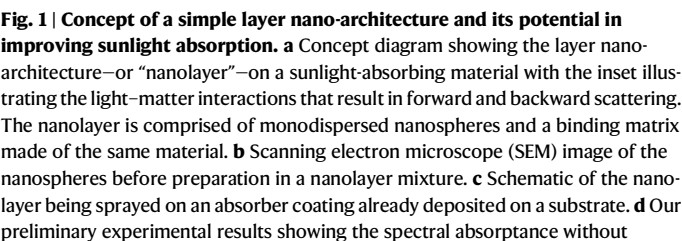

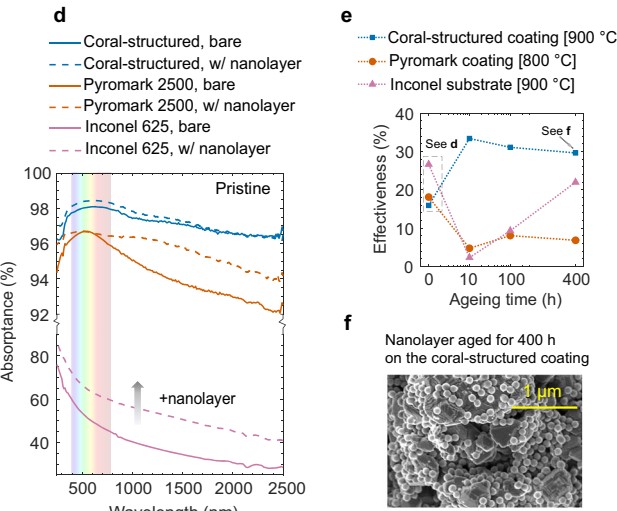

**Fig. 1 | Concept of a simple layer nano-architecture and its potential in improving sunlight absorption. a** Concept diagram showing the layer nano-architecture−or "nanolayer"−on a sunlight-absorbing material with the inset illus-trating the light−matter interactions that result in forward and backward scattering. The nanolayer is comprised of monodispersed nanospheres and a binding matrix made of the same material. **b** Scanning electron microscope (SEM) image of the nanospheres before preparation in a nanolayer mixture. **c** Schematic of the nano-layer being sprayed on an absorber coating already deposited on a substrate. **d** Our preliminary experimental results showing the spectral absorptance without nanolayer (solid lines) and with nanolayer (dashed lines) on a coral-structured coating[16], a Pyromark 2500 coating and Inconel 625 substrate[37], all in pristine condition. **e** Nanolayer effectiveness in pristine condition and after ageing at 900 °C for the coral-structured coating and Inconel 625, and 800 °C for Pyromark; absorber coatings are deposited on Inconel. See Eq. (1) for the definition of "effectiveness" and Supplementary Fig. 14 for the spectral absorptance data. **f** SEM image of the nanolayer on the coral-structured coating after ageing for 400 h. Source data are provided as a Source Data file.

For example, an effectiveness of 100% means that the reflection loss has changed from the baseline value to that of a blackbody with perfect sunlight absorption, or $\rho_{nl} = 0$. Likewise, an effectiveness of 50% means that the reduction in reflection loss has improved halfway from the baseline value to that of the ideal absorber, or $\rho_{nl} = \rho_{sm}/2$. Figure 1e shows isothermal ageing results (≥ 800 °C, ≤ 400 h) for the effectiveness of the nanolayer on different underlying absorbers and Fig. 1f shows a scanning electron microscope (SEM) image of the nanolayer on one of those coatings after 400 h ageing. Here, the solar absorbers with and without the nanolayer are aged side-by-side in a furnace, and their solar absorptance is then measured at room temperature to compute the effectiveness using Eq. (1), i.e. the nanolayer is aged together with the underlying material (not deposited on aged materials). The effectiveness of the nanolayer on the coral-structured and Pyromark coatings is nearly identical in a pristine condition ($t = 0$), but largely differs after ageing at different temperatures. Spectral absorptance measurements (Fig. 1d) show that the nanolayer improves absorption on the coral-structured coating mostly in the visible wavelength range where the solar irradiance is maximum. The nanolayer also improves the absorptance of Inconel throughout the measured wavelength range (Fig. 1d) and ageing time (Fig. 1e). In contrast, the improvement on Pyromark is found mostly in the infrared range, although with a larger improvement than that of the coral-structured coating in the visible range. The effectiveness $e_{nl}$ not only considers the spectral solar irradiance and absorptance but also the relative improvement in solar absorptance compared with an ideal blackbody.

## Effect of nanospheres on solar absorptance: theoretical prediction

We conduct computational electromagnetics (CEM) modelling based on a finite-difference time-domain (FDTD) method for analysing the effect of the layer nano-architecture on light trapping. Prior to con-ducting a detailed parametric assessment, we verified and validated our FDTD model (Supplementary Note 1). Monodisperse nanospheres can be tuned to enhance the net solar absorptance when applied on

various sunlight-absorbing materials, as shown by the effectiveness simulation results in Fig. 2a, upper panel. The nanolayer on a cutting-edge solar absorber material is most effective with nanospheres of ca. 100 nm in diameter. Here, the solar absorber is mimicked by a dummy material with a refractive index relatively close to that of air (antire-flective) and a moderate extinction coefficient (Supplementary Fig. 1a). These properties were chosen to produce a spectral absorptance close to that of Pyromark (Supplementary Fig. 1b) despite not having any roughness. In contrast, when the underlying absorber is tungsten (optical properties in Supplementary Fig. 1c) or a multilayer absorber[40], the optimum nanolayer effectiveness occurs with nano-spheres of ca. 120 nm diameter. The difference in optimum size is due to both material properties and spectral response. Simulation results in the lower panel of Fig. 2a show that the dispersion configuration, i.e. uniform (as in most metasurfaces[41]) versus random (as in our proposed layer nano-architecture), does not affect the solar absorptance near its optimum solar-weighted value.

Polydispersity in the nanosphere arrangement has generally a marginal effect on the nanolayer effectiveness, as shown in Fig. 2b. A mild downward trend of effectiveness is observed when increasing the standard deviation from the monodisperse condition to a relative standard deviation ($\sigma/D_{ave} \times 100\%$, where $\sigma$ is the standard deviation of the particle size distribution and $D_{ave}$ is the mean nanosphere dia-meter). For the mean diameter close to the optimal size of 100 nm, the absorptance did not change significantly with polydispersity within a relative standard deviation of 10%. A condition close to a mono-disperse arrangement seems beneficial, which is confirmed for the dummy material when introduced as the underlying absorber (Sup-plementary Fig. 2b, c shows results for other solar absorbers). This slightly downward trend is due to a reduced resonance effect (lower absorption peak) for polydisperse nanospheres compared with a monodisperse distribution (Supplementary Fig. 2a shows spectral absorptance). In some cases, a slightly higher effectiveness is observed for nanospheres with mild polydispersity (<5% in relative standard deviation) but the change is marginal, i.e. $e_{nl} < 1\%$. Therefore, the simulations that follow in this study consider monodisperse

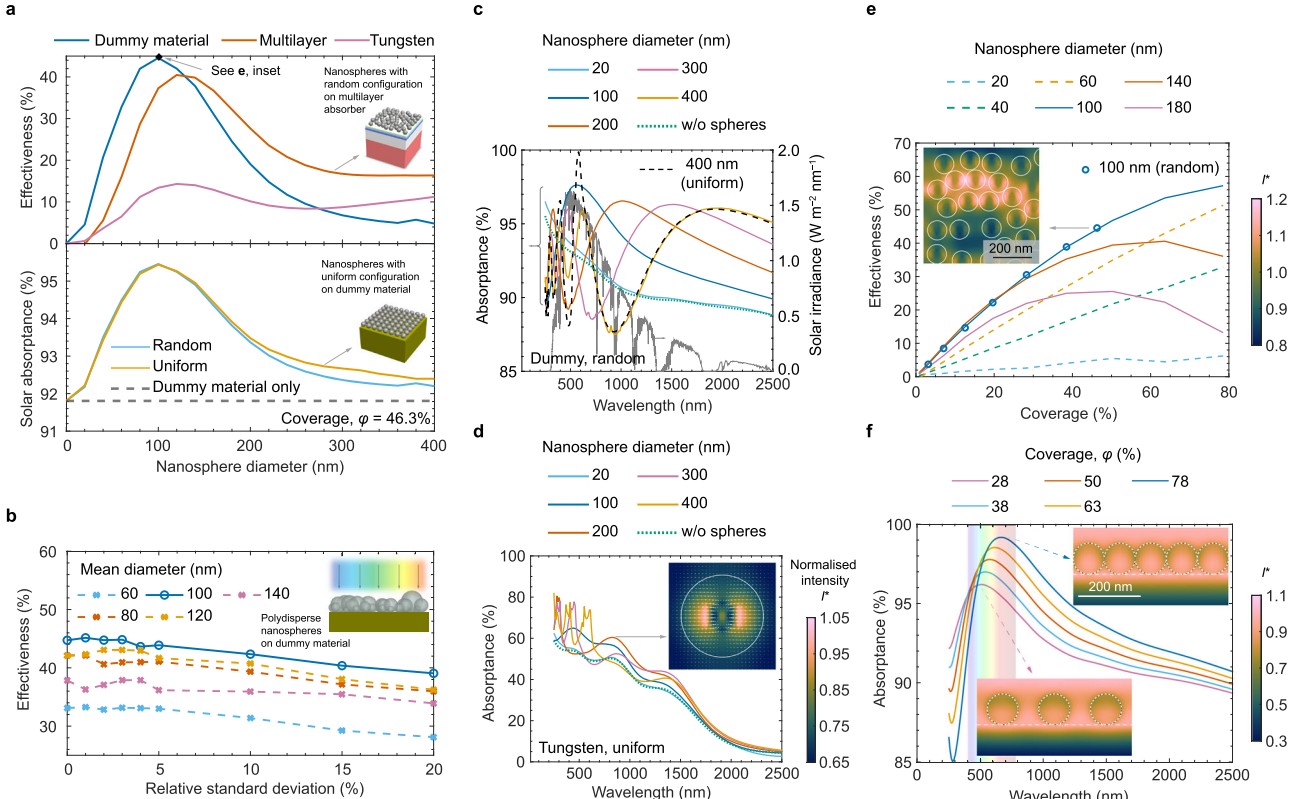

**Fig. 2 | Simulation results showing the effect of the nanosphere size and coverage on light absorption. a** The effect of the nanoshpere size (without matrix) for a surface coverage of 46%. Upper panel: Effectiveness of a random array of nanospheres when deposited on three different underlying sunlight absorbers. The inset shows the nanospheres on a multilayer composite, the former being tangent to the outermost layer. Lower panel: Solar absorptance as a function of the nanosphere diameter when monodispersed on a dummy material and two dispersion configurations: random and uniform, the latter shown in the inset. **b** Effectiveness as a function of polydispersity, expressed as the relative standard deviation in nanosphere size. The nanolayer is deposited on the dummy material for five mean nanosphere diameters, which are close to the optimal monodisperse case. The inset shows the side view when $D_{ave} = 100$ nm and relative standard deviation of 20%. The relative standard deviation is the standard deviation normalised against its mean value and expressed in percentage. **c** Spectral absorptance for monodispersed nanospheres having five different diameters when deposited on a dummy material with random configuration, and a uniform configuration is included in as a reference. **d** Spectral absorptance for monodispersed nanospheres when deposited on tungsten with uniform configuration, where the inset shows the energy distribution just under a nanosphere with 200 nm in diameter at a wavelength of 820 nm. **e** The effectiveness with uniform configuration as a function of the coverage for different diameters on dummy material. The markers show the effectiveness for a random configuration with a diameter of 100 nm. The inset shows the normalised intensity for light with the wavelength of 560 nm just underneath the nanospheres with a coverage of 46%. **f** Spectral absorptance of the absorbing material covered with 100 nm nanospheres with uniform configuration for different coverage. The insets show the magnitude of the normalised light intensity from a vertical plane cross-sectioning the nanospheres at a wavelength and coverage ratio of 520 nm and 28.3%, respectively (bottom), as well as 670 nm and 78.5% (top). Source data are provided as a Source Data file.

nanospheres. We found that, throughout most of the spectrum, the spectral response largely depends on the nanosphere diameter, as shown in Fig. 2c, d where primary and secondary solar absorptance peaks (at longer and shorter wavelengths, respectively) are observed for the nanolayer on the dummy absorber and tungsten. A uniform configuration does exhibit a significantly larger secondary absorptance peak compared to the random configuration, due to a resonance effect, but in a narrow band gap. As the nanosphere diameter is increased, the absorption peaks shift towards longer wavelengths.

Secondary absorption peaks transpire at shorter wavelengths. For example, the peak for the nanosphere array of 400 nm in diameter when placed on the dummy material occurs at the wavelength of ca. 650 nm (shown by the yellow solid line in Fig. 2c). This high absorptance coincides with the wavelength range of high solar irradiance. However, secondary peaks occur in a narrow wavelength band (Fig. 2c, d and Supplementary Fig. 3b, c), which greatly limits their contribution to increasing the solar absorptance (Fig. 2a, lower panel). Furthermore, these minor peaks shift towards longer wavelengths with increased nanosphere diameter, the same trend as the primary absorption peaks. In addition, the magnitude and width of those absorption peaks are largely dependent on the optical properties of the underlying material,

as shown in Fig. 2c, d for the dummy material and tungsten, respectively (see Supplementary Fig. 3 for a multilayer absorber).

As mentioned above, Fig. 2a (lower panel) shows that there is a marginal difference in solar absorptance between uniform and random arrangements of nanospheres. Therefore, a uniform configuration can be used in an initial theoretical optimisation (also shown in Supplementary Fig. 3a). Note that when our nanolayer is sprayed on an absorber material (Fig. 1c) a random distribution of the nanospheres, not uniform, is obtained (e.g. Fig. 1f). Although modelling such a random distribution is possible, it is computationally expensive to run large-scale multi-parameter simulations. Therefore, a uniform configuration is initially implemented in the parametric simulations and analyses in this work. For the optimal nanosphere diameter between 100 nm and 120 nm, the secondary peak forms at the ultraviolet (UV) range where the solar irradiance remains low, so its impact on the solar absorptance is negligible.

The portion of the surface area of the underlying solar material covered by the nanospheres (here termed "coverage" $\varphi$, based on the vertical nanosphere projection onto the underlying planar surface) impacts the effectiveness, as shown in Fig. 2e, f. A coverage greater than 60% for nanospheres of 100 nm (Fig. 2e) yields an effectiveness

greater than 50% for the dummy material. We found that the effectiveness monotonically increases with coverage for nanospheres smaller than 120 nm, while the effectiveness exhibits a local maxima for nanospheres larger or equal to 140 nm (detailed solar absorptance data in Supplementary Fig. 4c). This trend can be explained by the spectral response for different coverage, as previously discussed for variable nanosphere size (Fig. 2c, d). For nanospheres with 100 nm in diameter taken as the example (Fig. 2e), the larger the coverage the greater the increase of spectral absorptance in the visible range explaining its monotonic increment. However, the absorption peak shifts away from the solar irradiance peak in green light (Fig. 2f) explaining the tendency of the effectiveness to plateau. The magnitude of the absorption peak is reflected by the normalised cross-sectional energy distribution at the corresponding wavelength (insets in Fig. 2f). The closely packed nanospheres accumulate electromagnetic energy that is absorbed within the structure, both underlying solar material and nanospheres, while there is less electromagnetic energy captured when there is more spacing between nanospheres (i.e. smaller coverage). The absorption peak of nanospheres with a low coverage normally occurs near the UV range, shifting to the visible range as the coverage is increased.

It is worth noting that, for the same number of nanospheres, the polydisperse distribution would statistically introduce a higher coverage than the corresponding monodisperse case with the same mean particle size. However, we found that an increased degree of polydispersity (measured by the relative standard deviation) does not affect the wavelength at which there is a peak in spectral absorptance (Supplementary Fig. 2a), yet the absorption peak slightly drops. In addition, the potential irregular shape of nanoparticles is also considered by assuming an ellipsoidal particle shape. Interestingly, the

results of the effectiveness and spectral absorptance for monodisperse ellipsoids (with uniform configuration, Supplementary Fig. 5) are similar to the monodisperse nanospheres with variable coverage shown in Fig. 2e, f, suggesting a correlation between ellipsoid parameters and coverage. Furthermore, the packing configuration of uniform versus random yields significantly different energy distributions, as shown in the insets of Fig. 2d (uniform) and Fig. 2e (random, see also Supplementary Fig. 4b), despite having nearly identical effectiveness for nanospheres smaller than 200 nm, as shown in the lower panel of Fig. 2a. The normalised light intensity just underneath the nanospheres with random configuration (coverage of 46% and wavelength of 560 nm) indicates a largely heterogeneous light intensity distribution. Moreover, a boost in energy density is found between closely packed nanospheres when aligned in the same orientation as the polarisation direction of the incident light (more detailed studies about the influence of linear polarisation can be found in Supplementary Fig. 6).

### Effect of matrix on solar absorptance: theoretical prediction

Nanospheres adhere to the underlying solar material with a matrix, which also affects the light–nanolayer interaction. For example, during the nanolayer deposition onto a coral-structured absorber coating, the mixture of the organic solvent with nanoparticles is sprayed producing a binding matrix[16]. Without the presence of nanospheres, the matrix itself acts as an antireflective layer whose effectiveness depends mostly on its thickness and optical properties relative to the adjacent materials (air and underlying absorber)[42]. While different underlying absorber materials affect the effectiveness of the matrix, a relatively thick matrix between 50 nm and 100 nm and without nanospheres can yield a significant improvement in effectiveness, above 50% for both the dummy material (Fig. 3a) and a multilayer absorber

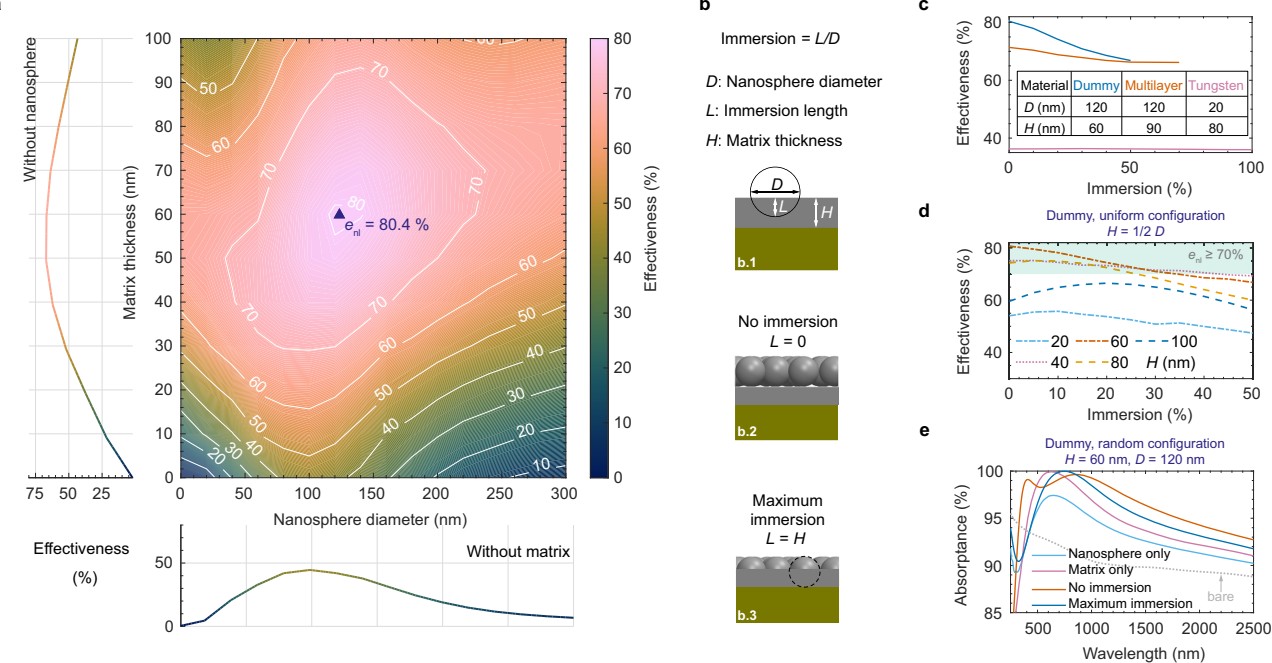

**Fig. 3 | Simulation results showing the effect of matrix thickness and immersion on light absorption. a** Nanolayer effectiveness as a function of nanosphere diameter and matrix thickness for monodispersed nanospheres with random configuration placed on the matrix (without immersion). Here, the underlying sunlight absorber is the dummy material. The adjacent two-dimensional plots show the effectiveness when having nanospheres only (without matrix; bottom) and matrix only (without nanospheres; left). **b** Sketch illustrating the definition of immersion (b.1) and two special cases: without immersion (b.2) and maximum immersion (b.3), i.e. when the nanospheres lay on the matrix and substrate,

respectively. **c** Effectiveness of the nanolayer with random configuration as a function of immersion for three different underlying absorbers. The nanosphere diameter and matrix thickness are fixed to those values that yield a maximum effectiveness when there is no immersion (listed in the inset table). **d** Nanolayer effectiveness as a function of immersion when the layer is placed with a uniform nanosphere configuration on the dummy material. The matrix thickness $H$ is half the nanosphere diameter $D$. **e** Spectral absorptance of the dummy absorber having the nanolayer with random nanosphere configuration and different morphological considerations. Source data are provided as a Source Data file.

(Supplementary Fig. 7b). Therefore, the adhesive matrix by itself has additional benefits in light trapping enhancement. When both matrix and nanospheres are combined in our layer nano-architecture, an even higher optimum effectiveness can be achieved, i.e. 80.4% for the dummy material and 71.4% for the multilayer absorber. A further extended simulation regarding different materials of nanolayer (both nanosphere and matrix) is conducted. Although materials with a higher refractive index (e.g. $\alpha-Al_2O_3$) may introduce more reflectivity when used as a layer (matrix) without nanospheres, a larger scattering by the nanospheres may occur (based on Mie theory). Thus, a slightly greater effectiveness is observed for alumina (Supplementary Fig. 8a) than silica (Fig. 3a) when deposited on the dummy material. However, a nanolayer with a larger refractive index than that of underlying material (e.g. rutile $TiO_2$ *vs.* dummy material) may increase the reflectance ($e_{nl} < 0$, Supplementary Fig. 8b) due to the occurrence of total internal reflection for scattered light with an angle of incidence onto the underlying material greater than a critical value. Silica seems to be a suitable choice for the nanomaterial because of both its broad range improvement in effectiveness (always $e_{nl} > 0$ for the studied parameter range, Fig. 3a, Supplementary Fig. 7) and its industrial scale production.

The above simulations consider a nanolayer with spheres placed on top of the matrix without immersion, i.e. the nanospheres are tangent to the original matrix–air boundary. However in a real spray deposition, nanospheres should be immersed in the matrix to produce a cohesive nano-architecture. The nanosphere "immersion"–defined in Fig. 3b–quantifies how much the nanospheres penetrate into the matrix, while the maximum immersion may be limited by the matrix thickness and nanosphere size. Figure 3c shows the influence of immersion on the effectiveness when the nanosphere diameter and matrix thickness are fixed to those values that yield a maximum effectiveness in the case without immersion (conditions listed in the inset table). These results show that the immersion decreases the effectiveness when optimum parameters without immersion are first fixed. In contrast, when an arbitrary nanosphere diameter and matrix thickness are chosen (excluding their optimum values without immersion), the immersion can improve the effectiveness, e.g. as shown in Fig. 3d for nanosphere diameter of $D = 200$ nm (matrix thickness fixed to $H = D/2 = 100$ nm). For a near-optimum nanosphere size of $D = 120$ nm ($H = 60$ nm), an immersionof about 20% yields a realistic and robust nano-architecture with nanolayer effectiveness greater than 70%. This is a promising theoretical prediction for using the proposed nano-architecture as a pathway towards achieving near-blackbody solar absorption.

The spectral absorptance has a strong dependence on the presence of nanolayer elements (nanospheres and matrix) and the nanosphere immersion. Figure 3e shows a case whereby the optimum nanosphere diameter and matrix thickness found for the case without immersion were set constant (matrix thickness of 60 nm and nanosphere diameter of 120 nm) with the dummy material as underlying solar absorber (see Fig. 3c). The shift in spectral absorptance with its peak shifting from green light ("matrix only" curve) to red light ("maximum immersion" curve) within the visible range explains the decrease in effectiveness with increased immersion. The optimum effectiveness is achieved when the wavelength of the absorption peak coincides with that of green light, i.e. the wavelength corresponding to the maximum solar irradiance. Likewise, the increase in effectiveness by increased immersion (without optimum nanosphere size and matrix thickness) is a consequence of the absorption peak shifting to green light.

## Enhancement of solar absorptance by nanolayer: experimental confirmation

CEM simulation results show that the proposed nanolayer can improve the absorptance of solar absorbers. The improvements are strongly impacted by the parameters of the nano-architecture and underlying material. Here, we conduct experiments with a dual purpose: (1) to find experimental evidence that a nanolayer comprised of randomly monodispersed nanospheres and a matrix improve the solar absorptance when applied on an arbitrary underlying solar material, and (2) to investigate the durability of the nanolayer after extensive high-temperature thermal ageing, for which no accurate theoretical framework has been developed.

Figure 4a plots spectral absorptance measurements of a coral-structured coating[16] with a nanolayer deposited on its surface for different nanosphere diameters, in pristine condition (see Supplementary Fig. 13 for other coverages). These experimental results show a good qualitative agreement with our theoretical predictions (Fig. 2a), i.e. the absorptance increases with nanosphere diameter from 12 nm until a near-optimal improvement with nanospheres of ca. 117 nm in diameter (Supplementary Fig. 9e shows the measured size distribution). Likewise, the experimental results in Fig. 4b show that an increased coverage from 10.6% to 60.7% yields larger solar absorptance, which is again in agreement with our theoretical predictions (Fig. 2e). The coverage can be tuned experimentally by varying the concentration of nanospheres in our nanolayer mixture, e.g. 0.215% silica nanosphere concentration in the nanolayer mixture (pre-deposition) yields a coverage of 30.7%, or by increasing the number of spray depositions. The measurements in Fig. 4b show that the nanolayer improves the absorptance in a wide spectrum from the visible to the near-infrared wavelength ranges. Note that there are differences in spectral absorption peaks for varying nanosphere diameter and coverage compared with theoretical predictions (e.g. Fig. 2c, d, f). This is because the optical properties of the coral-structured coating are different to those of the simulated materials (i.e. dummy absorber, tungsten and multilayer).

Figure 4c (upper panel) shows the solar absorptance for different coverage as a function of isothermal ageing time up to 1000 h at 900 °C. Interestingly after ageing, the nanolayer is most effective when having a coverage of 30.7% not the higher value of 67.4%. The effectiveness as a function of ageing time is shown in Fig. 4c (lower panel) for the best coverage. After extensive ageing for 1000 h at 900 °C, the importance of a moderate coverage of 30.7% is more pronounced, as shown in Fig. 4d, whereas in pristine condition a coverage between 30% and 70% yields nearly similar effectiveness. The nanolayer improves the solar absorptance by more than 1% (absolute value) after ageing, while the improvement is ca. 0.4% in the pristine condition, compared to the corresponding cases without the nanolayer. As the baseline absorptance is already large (i.e. exceeding 96%), this seemingly marginal improvement in solar absorptance is actually significant because the measured effectiveness exceeds 30%. Importantly, the solar absorptance of the coating with nanolayer drops less after the initial 10 h of ageing than that of the coating without nanolayer, i.e. the nano-architecture introduces an optical resilience not found in the coating without nanolayer. Crucially, the sample with a nanolayer retains its improvement in absorptance even after extensive high-temperature thermal ageing. This improvement and optical resilience are essential features needed in the CST industry because degradation of solar materials always occurs due to phase changes[13] and grain growth[27].

The stability of the nanolayer material is further confirmed by the SEM analysis in Fig. 4e. In pristine condition after nanolayer deposition, a large number of silica nanospheres with diameter of 117 nm covered the coating surface. The nanosphere diameter is dictated by the original nanosphere size (before adding to the nanolayer mixture) and the thickness of matrix, which covers the nanospheres. Minor morphology changes are observed in the nanospheres after ageing at 900 °C for 100 h, 400 h (Supplementary Fig. 10) and 1000 h (Fig. 4e), regardless of the coverage. However, the effectiveness as a function of coverage reported in Fig. 4d suggests that the nanolayer with the

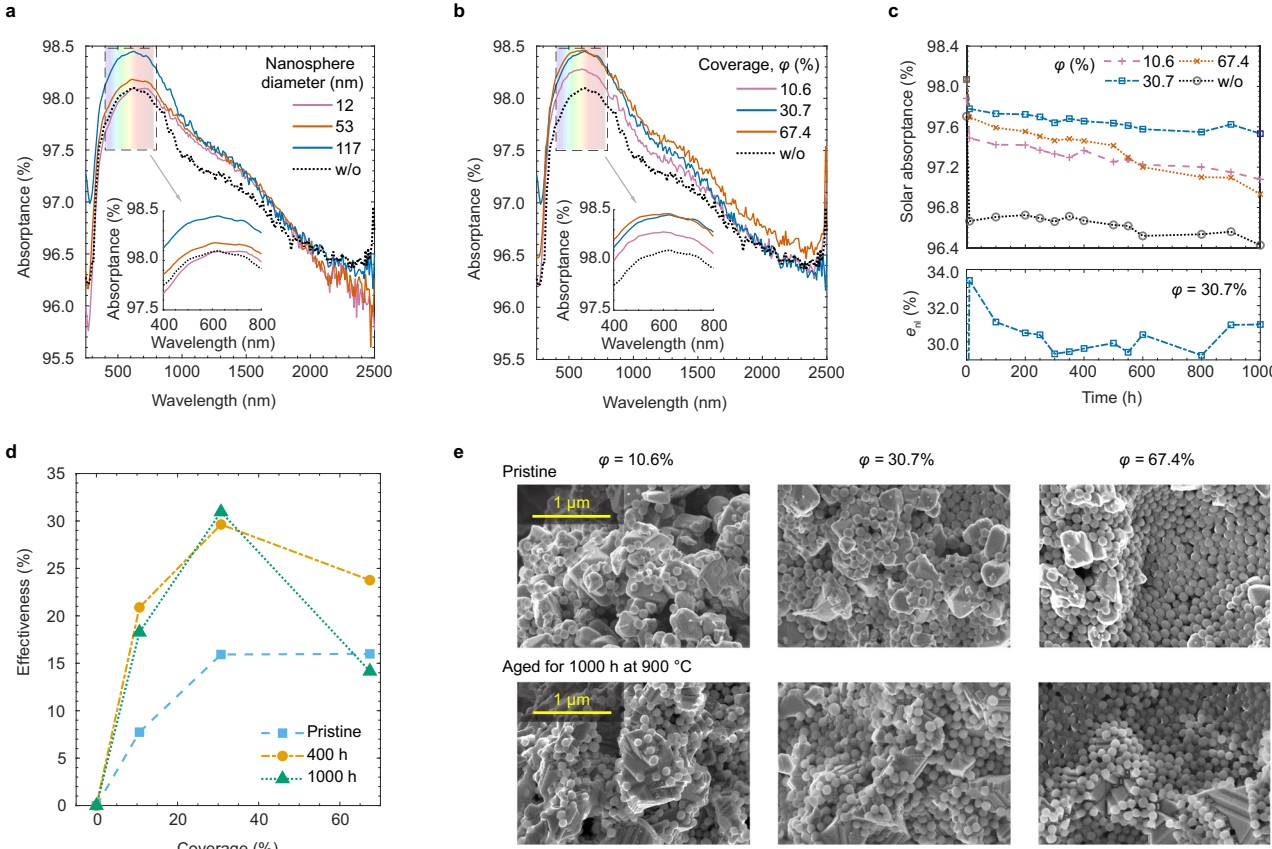

**Fig. 4 | Experimental evidence of optimal nanolayer effectiveness when applied on a coral-structured coating. a** Spectral absorptance of the coral-structured coating, one of the best available high-temperature sunlight absorbers[16], without nanolayer and with nanolayer having three nanosphere diameters with coverage of 30.7%. The real nanosphere diameter is generally larger than the nominal value because the matrix covers the nanosphere. **b** Spectral absorptance of the coral-structured coating without nanolayer and with a nanolayer having different coverage, all with nanosphere diameter of 117 nm. **c** Solar absorptance as a function of ageing time for the coral-structured coating without a nanolayer (baseline), and with nanolayer of different coverage (indicated in the legend) all having a nanosphere diameter of 117 nm, for an ageing temperature of 900 °C. The bottom plot shows the best-resulting nanolayer effectiveness for the coverage of 30.7%. **d** Effectiveness as a function of coverage for pristine condition and after ageing at 900 °C for 400 h and 1000 h. **e** SEM analysis of the nanolayer with different coverage on the coral-structured coating. The SEM images are taken after nanolayer deposition and before ageing (pristine condition) and after ageing for 1000 h. Source data are provided as a Source Data file.

largest coverage is far from optimum after extensive high-temperature ageing, despite exhibiting one of the highest effectiveness in pristine condition. It is worth noting that prior to deposition, the nanoparticles are well dispersed within an organic solution containing a silica precursor that forms the matrix after the nanolayer mixture is deposited (see Methods for details). This mixture preparation prevents the agglomeration of nanospheres and thus, after deposition, it lowers the likelihood of nanosphere sintering when exposed to high temperatures for a long time[27]. Furthermore, the solar absorber materials that our nanolayer aims to enhance are all operated well below the glass transition temperature of pure silica (ca. 1200 °C). For example, the surface temperature range of a conventional CST receiver generally falls between 600 °C and 700 °C. In fact, we use a higher temperature (900 °C) in our experiments to accelerate the kinetics and shorten the ageing time. Nonetheless, morphological changes in the underlying solar absorber such as crystal grain growth could have influenced the absorptance in ways not captured by our static electromagnetics modelling. Further work on the dynamic modelling of light–matter interaction in time-dependent nanoscale morphologies is needed to better understand these experimental observations.

After experimentally finding a near optimum effectiveness of the nanolayer obtained with nanospheres of ca. 117 nm and coverage of ca. 30.7%, we note that the matrix thickness is very thin, i.e. less than 10 nm. Based on our theoretical predictions in Fig. 3, another potential

improvement is that of a nanolayer with thicker matrix and moderate immersion. We then designed a nanolayer with thicker matrix by increasing the mass ratio of silicon in the organic liquid precursor to that in the nanospheres. Furthermore, to enable spray deposition on heated underlying materials (or "hot deposition"), we modified the nanolayer mixture using polymerised silica precursor instead of a monomer (tetraethyl orthosilicate). In our previous work[16], we used unpolymerised silica as precursor for deposition at room temperature (or "cold deposition"), but found that the matrix can crystallise before reaching a heated substrate producing a weak adhesion. Moreover, the organic silica precursor (monomer) may evaporate together with the organic material before crystallisation, which may be a problem for high-temperature in situ deposition. In contrast, the polymer silica evaporates slowly leaving a thicker matrix. Deposition on heated substrates can improve thermal stability at higher temperatures (due to residual thermal stresses), as well as enabling in situ refurbishment and maintenance of CST coatings under operation.

Figure 5a shows a better solar absorption of the coral-structured coating when having such a thick-matrix nanolayer (with polymerised precursor and hot deposition) both in pristine condition and after extensive thermal ageing at 900 °C, compared with the thin-matrix nanolayer (with a monomer precursor and cold deposition). Our previous work is also included for comparison (dotted line)[16] where the nanolayer was not optimised by tuning matrix properties (thickness

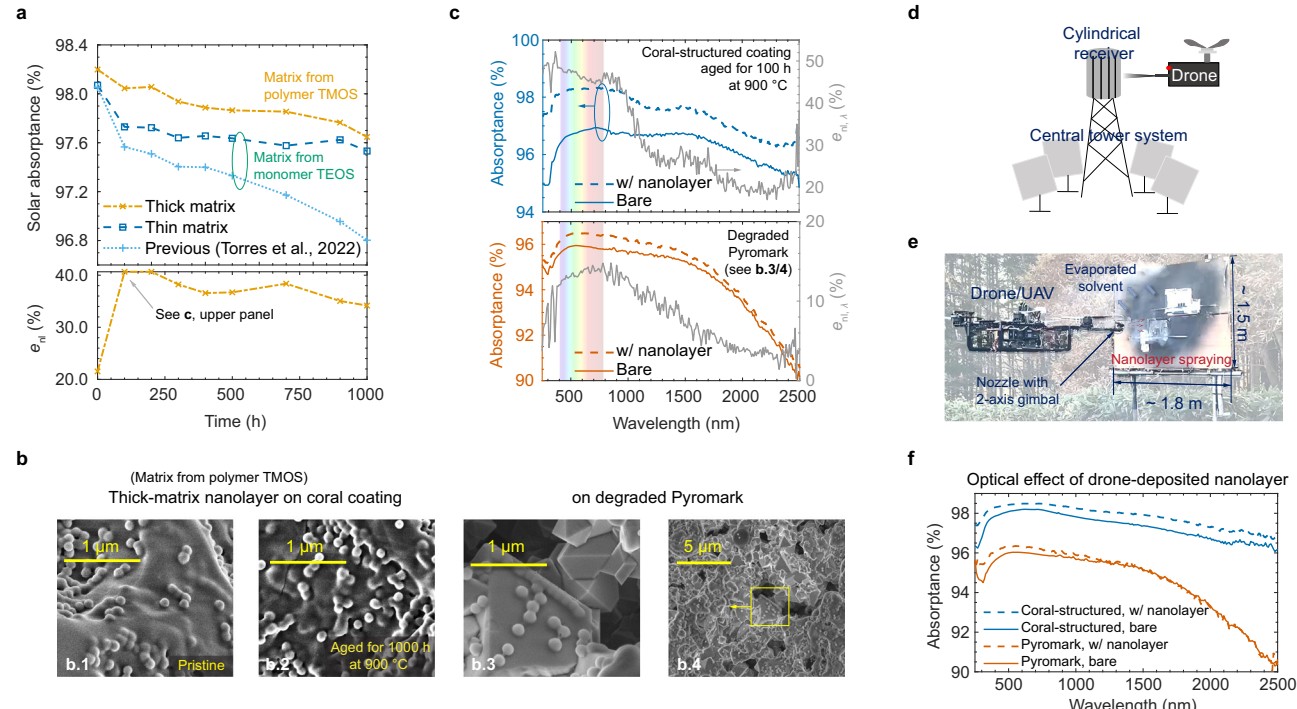

**Fig. 5 | Characterisation and scalability of improved nanolayer. a** Solar absorptance as a function of time when aged at 900 °C for a nanolayer having thick matrix with polymerised silica precursor or thin matrix with unpolymerised silica precursor (same coverage of 30.7%, as in Fig. 4c). The reference coating[16] claimed a record of solar absorptance at high temperatures. The effectiveness of our best-performing nanolayer is shown in the lower plot. **b** SEM analysis of nanolayers with thick matrix on the coral-structured coating in pristine condition and after ageing for 1000 h and nanolayers with thick matrix on Pyromark cured at 950 °C (degraded Pyromark) in different magnifications. **c** Spectral absorptance without nanolayer (solid lines) and with the improved nanolayer having a thick matrix

(dashed lines) on the coral-structured coating (upper panel) at 900 °C after ageing for 100 h and degraded Pyromark (lower panel). The spectral effectiveness in the lower plot shows a 50% effectiveness in the visible range for the coral-structured coating. **d** Scalability concept for large-scale nanolayer deposition using a drone. The nanolayer can be sprayed on the surface of a central tower solar thermal receiver. **e** Photo of our drone spray testing in an outdoor environment. **f** Spectral absorptance of the coral-structured coating and Pyromark coating in pristine condition having the nanolayer deposited by a drone. Source data are provided as a Source Data file.

and precursor composition). Importantly, the thick layer nano-architecture produces a stable solar absorptance of $97.88 \pm 0.14\%$ (temporal average ± standard deviation) between 100 and 1000 h of ageing at 900 °C. Furthermore, a large solar absorptance of 97.64% is maintained and the effectiveness is greater than 35% after ageing for 1000 h. The experimental effectiveness reaches values as high as 40% after ageing for 100 h (Fig. 5a, lower panel), yet theoretical predictions in the improvement of effectiveness (Fig. 3) suggest that this nanolayer structure can be further improved.

Compared to our previous study[16], the two nanolayers produced in this work ("thin" and "thick") exhibit improvements in solar absorptance and stability. This improvement is partly due to slight modifications to the underlying coral-structured morphology (see Methods), but mostly due to the increased matrix thickness and hot deposition enabled by the polymerised silica precursor. The thicker matrix can be seen in the SEM analysis of Fig. 5b. 1,2 (compared to Fig. 1f). The silica produced from the unpolymerised and polymerised precursors may yield a different optical property (complex refractive index; further confirmation is needed). The experimental observation of a thicker matrix with immersion being beneficial agrees with the theoretical predictions where the nanosphere diameter and matrix thickness are not optimised for the case without immersion (dark blue dashed line in Fig. 3d). More accurate control of matrix thickness and immersion is needed with the proposed scalable deposition method to better optimise solar absorptance based on theoretical predictions. In addition, negligible changes in both matrix and nanosphere morphologies after extensive thermal ageing corroborate the durability of the nanolayer.

Figure 5c shows the spectral absorptance of two types of solar absorber coating without a nanolayer (i.e. solid lines indicating a bare coating) and with a nanolayer (i.e. dashed lines). An enhancement is achieved for both the coral-structured coating and degraded Pyromark, yet the former exhibits a more pronounced improvement in the visible range. Note that the Pyromark coating is prepared with different conditions (cured at 950 °C, see Methods) to the Pyromark shown in Fig. 1d (cured at 500 °C[37]). Curing Pyromark at the elevated temperature of 950 °C yields a lower initial solar absorptance but a better optical and mechanical stability during the ageing[27,43], compared with Pyromark cured at a lower temperature[18,44]. In addition, a similar morphology was observed between pristine Pyromark cured at a high temperature and an aged Pyromark sample originally cured at a lower temperature[19]. For example, the degraded Pyromark exhibits a similar crystalline morphology (Fig. 5b. 3, 4) as a Pyromark coating cured at 750 °C and then isothermally aged for 3000 h at 800 °C[19]. The higher curing temperature then allows us to model degraded Pyromark that has not yet failed.

The spectral improvement in solar absorption is quantified by the spectral nanolayer effectiveness defined as

$$e_{nl,\lambda} = \left(1 - \frac{\rho_{nl,\lambda}}{\rho_{sm,\lambda}}\right) \times 100\%. \tag{2}$$

A spectral effectiveness of nearly 50% is achieved throughout the entire visible range for the coral-structured coating (Fig. 5c, upper panel), which demonstrates a substantial light-trapping enhancement by the layer nano-architecture. A significant spectral absorptance

improvement for Pyromark (Fig. 5c, lower panel) further demonstrates that our nanolayer can be used as means of maintenance of degraded solar absorbers, including those widely used in the CST industry. SEM analysis confirms the excellent stability and adhesion of the nanolayer on both the coral-structured and degraded Pyromark coatings (Fig. 5b). Furthermore, Fig. 5d, e show the concept of spraying the improved nanolayer (with a polymerised silica matrix) using a drone and the scalability testing on heated samples. The resulting improvement in the pristine condition is shown in Fig. 5f, noting that each curve is the average spectral absorptance of four samples, with ± 0.05% variation and confidence level of 90%. This further confirms the suitability and scalability of our nanolayer as an upgrading, refurbishment or maintenance process with semi-automated in situ re-painting for pristine or degraded solar absorbers (see Supplementary Note 2 and Supplementary Movie 1 for more details on the nanolayer scalability experiments using a drone).

It is worth noting that a nanolayer-based improvement in solar absorptance (from a baseline solar absorptance of ca. 97%) is expected to increase the thermal emittance due to Kirchhoff's law, which states an equivalence of spectral emittance and spectral absorptance in thermal equilibrium. An increase in thermal emittance on high-temperature surfaces results in a greater radiative heat loss due to the Stefan–Boltzmann law. However, most central tower CST applications have sunlight concentrations exceeding 1000 suns, meaning that the negative impact on receiver efficiency from the high emittance is much smaller than the positive effect from a high solar absorptance[45]. In addition, nanoscale features in our nanolayer (ca. 120 nm) are much smaller than the dominant wavelength of the thermal emission (ca. 3000 nm for a blackbody at 700 °C, based on the operating temperature of CST receivers). Therefore, nanoscale features cannot be tuned to effectively control thermal emission in the infrared range. We measured the relative increment of emittance at high temperatures (Supplementary Note 3) and confirmed an increase of ca. 1.4%. However, its impact on photo-thermal energy conversion is negligible (as assessed by a *figure of merit*, Supplementary Table 1) when deposited on a solar absorber coating exposed to sunlight concentration of 1000 suns.

## Discussion

This paper provides theoretical and experimental insight into the light–matter interaction and resulting solar absorptance improvement when applying a layer nano-architecture onto an existing solar absorber material. Our proposed nano-architecture is comprised of randomly monodispersed nanospheres and a binding matrix, and can be applied onto any type of sunlight-absorbing material including existing (e.g. commercial) and future solar absorbers. Electromagnetics modelling shows that a wide-spectrum light trapping enhancement can be achieved for a variety of underlying solar absorbers, such as tungsten and multilayer coatings. The performance is found to largely depend on the optical properties of the underlying material and nanolayer parameters such as nanosphere diameter, coverage, matrix thickness and immersion. Monodispersed nanospheres with random and uniform configurations exhibit comparable solar absorptance. Theoretical predictions on a cutting-edge solar absorber suggest that we can achieve a nanolayer effectiveness greater than 70% with a robust nano-architecture (having a nanosphere immersion of ca. 20%).

The experimental results of the nanolayer on a solar absorber material match qualitatively with our theoretical predictions. We characterised the effects of the proposed nano-architecture on a coral-structured coating, and found that the nanolayer effectiveness can exceed 40%, i.e. the relative percentage value closer to the ideal blackbody solar absorption from the baseline absorptance without nano-architecture. A large solar absorptance of 97.64% is maintained, even after 1000 h of ageing at 900 °C. To the best of our knowledge, this is the largest absorptance reported under this extreme isothermal

ageing condition in air. The nanolayer also exhibits excellent long-time optical stability at this ageing temperature i.e. solar absorptance of 97.88 ± 0.14% (temporal average ± standard deviation) between 100 and 1000 h of ageing at 900 °C, with its effectiveness always exceeding 35%. The lack of significant morphological changes in the nanolayer is in agreement with the observed stability of the enhanced solar absorptance.

We demonstrate the scalability of our layer nano-architecture by a facile drone-assisted deposition. The potential application for upgrading the optical performance (solar absorptance and stability) of existing solar absorbers without the need to develop entirely new sophisticated materials and light-trapping structures is a noteworthy innovation. The drone-assisted deposition could be an inexpensive in situ upgrading or maintenance method for pristine or degraded sunlight-absorbing materials already in use or future coatings in CST technologies.

## Methods

### Solar absorptance and solar reflectance

The standard metric of solar absorptance $\alpha$ was calculated using the ASTM G-173 standard for the spectral solar irradiance $G(\lambda)$ as follows:

$$\alpha = \frac{\int_{280 \, \text{nm}}^{2500 \, \text{nm}} \alpha(\lambda)G(\lambda) \, \mathrm{d}\lambda}{\int_{280 \, \text{nm}}^{2500 \, \text{nm}} G(\lambda) \, \mathrm{d}\lambda}, \qquad (3)$$

where $\alpha(\lambda)$ is the spectral directional absorptance and $\lambda$ is the wavelength. The integration was within the lower limit of 280 nm (i.e. the minimum wavelength of ASTM G-173 data) and upper limit of 2500 nm (i.e. the largest wavelength detected with our spectrophotometer). For a completely opaque surface, the reflectance $\rho$ (or absorptance $\alpha$) can be calculated by $1 - \alpha$ (or $1 - \rho$). Hence, we can simply obtain the solar reflectance as $\rho = 1 - \alpha$, or

$$\rho = \frac{\int_{280 \, \text{nm}}^{2500 \, \text{nm}} \rho(\lambda)G(\lambda) \, \mathrm{d}\lambda}{\int_{280 \, \text{nm}}^{2500 \, \text{nm}} G(\lambda) \, \mathrm{d}\lambda}, \qquad (4)$$

where $\rho(\lambda)$ is the spectral hemispherical reflectance at a near-normal angle of incidence, which was measured with a spectrophotometer (PerkinElmer UV/VIS/NIR Lambda 1050) at an angle of incidence of 8°. The coral-structured sunlight absorber coating has a strong diffuse reflectance with negligible change in reflectance between normal and 8° angle of incidence[16].

### Computational electromagnetics simulations

The magnitude and direction of the Poynting vector were analysed by finite-difference time-domain (FDTD) using Ansys Lumerical, a commercial photonics simulation software. The incident wave was set as a plane wave with normal incidence to the absorbing surface as shown in Supplementary Fig. 11a. The side view of the calculation setup shows the periodic boundary conditions set along $z$ on the lateral planes $xz$ and $yz$; the perfectly matched layer (PML) boundary condition was set at the bottom and top $xy$ plane boundaries. The electric and magnetic fields were calculated by preset frequencies (related to the wavelength from 250 nm to 2500 nm) at each time step. The total reflected and transmitted powers at each frequency were calculated by integrating the Poynting vector across the entire calculation domain, as means of quantifying the transmittance and reflectance through direct comparison (i.e. computing the ratio) with the incident power. As a negligible amount of transmitted power is absorbed on the bottom boundary (due to the enforced opaque condition), the absorptance of the material can be calculated via the equation $\alpha(\lambda) = 1 - \rho(\lambda)$. The

complex refractive indices were obtained from the literature for silica[46], tungsten[47], mimicked cutting-edge absorber (or "dummy material")[16] and multilayer materials[40]. The dummy material only has the optical property of a complex refractive index, and does not include texture, chemical and mechanical properties. We considered the dummy material in the form of a semi-infinite slab. The optical properties of the dummy material, tungsten and silica are shown in Supplementary Fig. 1 as a reference.

A normal (Gaussian) size distribution was used for the simulations with polydisperse nanospheres. Although a lognormal distribution is generally more suitable than a normal distribution for describing the polydisperse nanoparticles, our particle size characterisation (Supplementary Fig. 9d, e) indicates that nanolayers with nanospheres of mean diameter above 100 nm do follow a normal distribution. We note that distributions with particles smaller than 50 nm in mean diameter follow a lognormal distribution. In the modelling, to avoid potential negative diameters in a normal distribution, the minimum diameter was set to zero. Representative distributions of polydisperse nanospheres are shown in Supplementary Fig. 2a.

The different configurations were set in the FDTD simulation, as shown in the top view of Supplementary Fig. 12. The periodic boundary was set on side boundaries. The coverage $\varphi$ was calculated by the fraction of the total projected area covered by the nanospheres to the total area as

$$\varphi = \frac{N \pi D^2}{4A}, \tag{5}$$

where $D$ is the nanosphere diameter, $N$ is the number of nanospheres and $A$ is the total solar absorber area within one periodic calculation domain. When changing the nanosphere diameter, the calculation domain was adjusted accordingly to maintain the same coverage.

### Underlying solar absorber coatings in experiments

**Coral-structured coating**. The coral-structured coating[16] is a hierarchical solar absorber with distinct length scales in a wide spatial range. The most distinct features of this coating are their protrusions and micropores, both reassembling stony corals. The macro-scale protrusions are ca. 80 μm in size, which is a length scale not found in conventional CST absorbers such as Pyromark 2500. The micropores are ca. 3 μm in size. The underlying solar absorber has two layers: the base layer that improves adhesion with the metallic substrate, and the absorption layer that improves light trapping via multiple-reflections between macro-scale protrusions. Both layers have micropores while the absorption material is black spinel pigments, which are bonded by alumina in the base layer and titania in the absorption layer. The unique aspect of this coating is its spray-deposition on a heated substrate. The heated substrate has two roles: first, it evaporates the deposited paint which has a large solvent (acetylacetone and titania precursor) to spinel pigment ($Cu_{0.64}Cr_{1.51}Mn_{0.85}O_4$) mass ratio of 40:1; second, the high temperature is required for the oxides to form via thermal decomposition or pyrolysis. More details on the characteristics of this coating are found in Torres et al.[16]. The solar absorptance in pristine condition is above 97.6% and after extensive thermal annealing at 900 °C is ca. 96.4%.

The coral-structured coating is prepared based on two mixtures for the base and absorption layers. To produce the base layer mixture, an aluminium complex and isopropyl di glycol were mixed by stirring before adding and mixing with the black spinel pigments to form the base layer mixture with a catalyst as the adhesive. To produce the absorption layer mixture, a titanium precursor was first reacted with acetylacetone at room temperature, and then heated at 80 °C for 6 h, and then diluted with 2-propanol. The black pigments and N-methyl-2-pirrolidone were added and dispersed by ultrasonication for 30 min. To deposit the coating, the base solution was sprayed with an airbrush

for 4 s while heating the substrate at 270 °C; the absorption solution was sprayed multiple times onto the base layer (held at 320 °C) with a large spraying nozzle. The two-layer coral-structured coating was formed with a ca. 20 μm thick base layer and an absorption layer with a series of protrusions having a characteristic dimension (i.e. height and width) of ca. 80 m. The tuning methods for the macro-scale protrusions[16] include adjusting absorption layer mixture for the ratio of the titanium precursor to acetylacetone, spray liquid pressure, and substrate temperature. The substrates were chemically cleaned (same procedure as for Pyromark below) but not grit blasted.

**Pyromark 2500**. Pyromark is a silicone-based paint that is widely used as a high-temperature absorber coating in CST receivers. It has an easy deposition method on large surfaces and has a high solar absorptance (96–97%) in pristine condition. However, Pyromark exhibits optical and mechanical degradation during operation at high temperatures[19]. For example, in the case of the Solar One plant, Pyromark absorptance decreased by about 6% after four years of working at 500 °C[10]. This high optical degradation rate leads to increased maintenance costs, e.g. when repainting the receiver to reach the initial absorptance. Degradation issues have motivated many studies to develop a new coating with better optical durability[16,18].

A requirement to obtain robust Pyromark samples is the substrates treatment. First, Inconel 625 was grit blasted using 250 μm mesh of white aluminium oxide at a pressure of 0.62 MPa. Following the grit blasting, they underwent a chemical cleaning procedure. In this process, the substrates were immersed in tetrachloroethylene for 60 s, followed by a 30 s gentle surface scrub using lint-free wipes. The same process was then repeated using methyl ethyl ketone. In the next stage, after mixing Pyromark paint for about 20 min, it was sprayed onto the substrates using an airbrush gun (Artlogic AC330) at a pressure of around 0.28 MPa. For uniform paint layer, the airbrush gun was positioned about 10 cm above the samples. Each paint layer was allowed to dry for a period before applying the next layer. This process was repeated until the desired paint thickness of 20–30 μm. After 24 h of drying at room temperature, samples underwent a curing process within a programmable muffle furnace. During the curing stage, the samples were heated to temperatures: 120 °C for 2 h, 250 °C for 2 h, 540 °C for 1 h, and ultimately 950 °C for 1 h[43].

### Nanolayer materials preparation

**Materials for room-temperature deposition using *unpolymerised* silica precursor (thin matrix)**. A sprayable nanolayer mixture was prepared from two precursor formulas: Mixtures A and B. Mixture A was prepared by diluting 41.7 g of tetraethyl orthosilicate (TEOS) with 126.7 g of ethanol, stirring at room temperature at 500 rpm for 1 h. A separate solution was prepared by adding 14.8 g of hydroxyacetone, which is a catalyst for dehydrating and polymerising TEOS, to 126.7 g of ethanol with 18 g of ion-exchanged water. After ultrasonic irradiation for 10 min, the latter solution was stirred at room temperature at 500 rpm for 1 h. The two above solutions were mixed and stirred at 40 °C at 500 rpm for 48 h, then kept at 40 °C with thermostatic bath for 72 h. This method produced 330 g of Mixture A. A lower silicon content was achieved by further diluting with an additional 1870 g of ethanol.

Mixture B was prepared by diluting 20.8 g of TEOS with 63.4 g of ethanol, 7.4 g of hydroxyacetone, and 9 g of ion-exchanged water. Colloidal silica (ORGANOSILICASOL™, Nissan Chemical Industries, Ltd.) was added and ultrasonically irradiated for 10 min. To accelerate the reaction, the well-mixed solution was constantly stirred at 500 rpm for 48 h at 40 °C. Same as with Mixture A, after 72 h at 40 °C in a thermostatic bath, this method yields 120 g of Mixture B, with 5% of silica nanosphere concentration. By diluting different amounts of ethanol, final silica nanosphere concentrations were achieved, i.e. 0.34 wt%, 0.86 wt% and 1.72 wt%. Diluted Mixtures A and B were then mixed in a 3:1 ratio to form the final composition for the nanolayer mixture.

By changing the concentration of silica nanospheres in the mixture, the coverage can be tuned, as shown in the SEM images after deposition (Fig. 4e). The coverage of 30.7% was obtained from Mixture B with the final silica concentration of 0.86 wt%.

The nominal diameter of silica nanospheres in the colloidal mixture is specified by the manufacturer: IPA-ST-ZL (nominal nanosphere diameter of 80 nm), IPA-ST-L (45 nm nominal) and IPA-ST (12 nm nominal). The measured nanosphere diameter before mixing in the nanolayer mixture and after deposition differs (Supplementary Fig. 9). Although a direct thickness measurement of thin matrices is challenging, e.g. via cross-section SEM, an indirect measurement can be devised assuming a simplified nanolayer morphology (Supplementary Fig. 9b). Here, we subtracted the effective nanosphere diameter pre-mixing (without matrix) from the post-deposition effective diameter (Supplementary Fig. 9d, e). A uniform matrix thickness around the real nanosphere is assumed to estimate the matrix thickness. This method confirms that the TMOS-based matrix is thicker than the TEOS-based matrix. In addition, a different number density of silica nanospheres can be obtained by increasing or decreasing the number of spray depositions. The coverage after deposition was calculated using SEM images and Eq. (5), where $D$ is the average measured diameter of the nanospheres.

**Materials for deposition on heated substrate using *polymerised* silica precursor (thick matrix).** The improved thick matrix using polymerised silica nanolayer can be applied to a heated substrate. 700 g of ethanol with 99.5% purity was mixed with 100 g of ion-exchanged water and homogenised by ultrasonication for 10 min. In this solution, 10 g of a methyl silicate oligomer (MKCTM silicate, grade name MS56; manufactured by Mitsubishi Chemical Corporation, which is an oligomer formed by the partial hydrolysis of tetramethoxysilane, or TMOS), and 6 g of a commercially available colloidal silica (ORGANOSILICASOLTM IPA-ST-ZL, manufactured by Nissan Chemical) were added with continual stirring at 550 rpm for 1 h at room temperature. The mixture was then added with 1.63 g of acetic acid, followed by stirring at 40 °C for 48 h, at a mixing speed of 550 rpm. To have better adhesion on the Pyromark coating surface, the solution was diluted at 1:1 ratio with ethanol (for lab-based deposition) and IPA (for drone-based deposition).

All materials used to produce the nanolayer mixture come from proven industrial processes that are suitable for mass production. This includes the silica nanospheres, industrial ethanol and silica precursors (TEOS and TMOS). Furthermore, the nanolayer has much less mass than that the underlying solar absorber coating (Fig. 1a), so material costs are reduced. Importantly, we have shown theoretically that our layer nano-architecture is not very sensitive to its dimensions (Fig. 3a) such as its nanosphere size ($e_{nl} > 70\%$, 100–150 nm), matrix thickness ($e_{nl} > 70\%$, 30–90 nm), immersion ($e_{nl} > 70\%$ for immersion < 20%) and polydispersity (Fig. 2b). The lax requirement for precision in manufacturing is another contributing factor to lowering costs.

### Nanolayer deposition method
**Lab-based deposition.** The sprayable solution prepared by unpolymerised silica was applied to solar absorber coatings by spraying (using a Colani 2400 airbrush) and curing. The nanolayer mixture was applied with a spray pressure of 0.3 MPa and 20 cm from the solar absorber coating surface for 3 s at room temperature. After spray application, the coupon was cured 400 °C for 30 min. This spray and curing process is repeated twice.

The sprayable solution prepared by polymerised silica was applied to pre-heated solar absorber coatings by spraying using the same type of airbrush. The nanolayer mixture was applied on a coral-structured coating (held at 300 °C) with a spray pressure of 0.3 MPa and 20 cm from the coating surface for 4 s. This spray deposition was repeated twice. The diluted nanolayer mixture solution was applied on Pyromark (held at 310 °C) with a spray pressure of 0.25 MPa and 20 cm from the coating surface for 3 s. The same as with the coral-structured coating, this spray deposition was repeated twice.

**Drone-assisted deposition.** A hotplate with the dimensions of 25 cm × 25 cm heated to 270–350 °C was prepared to emulate a local receiver surface controlled at the desired temperature (e.g. by circulation of molten salt). A camera and spray nozzle specifically designed for mid-air spraying were attached to the drone. Painting was performed remotely from the ground. The drone was equipped with a front-facing rangefinder (distance sensor) to maintain a distance of ca. 50 cm from the target while in flight. The distance was measured from the tip of the nose-mounted spray nozzle to the target surface. The spray nozzle was equipped with a two-axis gimbal, and the flight control system performed tilt-angle stabilisation and spray-trajectory-angle-error correction. The spray pressure was 0.2 MPa. The nanolayer was deposited on the absorber surface at a temperature of ca. 270 °C. The mixture prepared by the polymerised silica precursor was applied on the coral-structured coating for 10 s and repeated twice. The diluted nanolayer mixture solution was applied Pyromark for 10 s and repeated twice.

### Ageing and characterisation
The isothermal ageing was conducted in a programmable muffle furnace (LABEC SF–13–SD). The heating rate was set to 3 °C min⁻¹ to increase the sample temperature from room temperature to the ageing temperature (800 or 900 °C). The isothermal ageing period was set to 100 h or 200 h. After the ageing process, the furnace cooled down to room temperature at the rate of ca. 4 °C min⁻¹. The characterisation includes spectral reflectance measurement described in Sec. "Solar absorptance and solar reflectance" and morphology analysis with a scanning electron microscope (Zeiss UltraPlus analytical FESEM). Nanosphere diameters were measured from SEM images.

### Data availability
Source data are provided with this paper.

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

## Acknowledgements

This research was supported by the Australian Research Council (ARC) Linkage Project (LP170101239, recipients J.C., J.F.T., W.L. and A.T.) funded by the Australian Government, and the New Energy and Industrial Technology Development Organization (NEDO) New Energy Venture Business Technology Innovation Program (P10020, recipient K.T.) funded by the Japanese Government. Funding from industry partners Vast Solar and Nano Frontier Technology is acknowledged. The microscope analysis was conducted in the Centre of Advanced Microscopy (CAM) at the Australian National University node of the Microscopy Australia.

## Author contributions

Conceptualisation: K.T., Y.M., J.F.T. Methodology: Y.G., K.T., S.H., Y.M., J.F.T. Investigation: Y.G., K.T., S.H., Y.M., A.T., J.C., W.L., J.F.T.

Visualisation: Y.G. Funding acquisition: K.T., A.T., J.C., W.L., J.F.T. Project administration: K.T., J.C., J.F.T. Supervision: A.T., J.C., W.L., J.F.T. Writing —original draft: Y.G., J.F.T. Writing—review & editing: Y.G., S.H., A.T., J.C., J.F.T.

## Competing interests

The authors K.T., J.F.T., Y.M., and Y.G. declare a financial competing interest. They are inventors on an international patent (PCT) application on the nanolayer invention, filed on 7 July 2022 in the name of Nano Frontier Technology Co., Ltd., The Australian National University, and Shinshu University. The application number for the that patent filing is PCT/JP2022/027000. The PCT application is currently in the stage prior to the national phase entry. The remaining authors declare no competing interests.
