## [Peer Review File · Nature Communications]

REVIEWER COMMENTS

Reviewer #1 (Remarks to the Author):

In the submitted manuscript entitled “Scalable Nano-architecture for stable near blackbody solar absorption at higher temperatures”, the authors explored the applicability of the layer nano-architecture on solar absorbers by electromagnetic modelling. Experimentally demonstrated by application on well-established solar absorber coatings like Pyromark & coral-based absorber coatings and enhances the solar absorptance to 35% and stable over 1000h at 900 °C in air. Successfully demonstrate the scalability of the coating application in real field conditions to improve and maintain the solar absorption property of the CST absorber. This manuscript can be accepted after the necessary modifications with answers to all the review comments. Here I am listing a few remarks on reviewing this manuscript:

The author provided a clear idea of the application of nanomaterials in solar absorbing materials in CST systems. However,

1. What was the thickness of the nano-particle layer applied on the base solar absorber layer estimated?
2. Since the developed material applies to CST applications, why was only the solar absorptance property analyzed alone? What about the emittance of the coatings?
3. How to do the coverage & particle size of the nano-spheres on the base absorber materials affect the enhancement of solar absorptance? How is 30% coverage optimized?
4. Does the wide distribution of Nano-spheres (11 – 117 nm) influence the enhancement of the solar absorptance in the experimental section, whereas, in modelling, nanosphere size is considered?
5. Provide legends in Fig S4 (b)

Reviewer #2 (Remarks to the Author):

The authors showed a scalable method using silica nanoparticle deposition (nanolayer) to enhance the solar absorptance of state of the art solar absorbing materials such as Pyromark. The work is generally interesting. While the principles reported here are not quite new (using light trapping effect of nanoparticles), the application of using this method for potentially large scale deposition on solar absorbing materials is novel. However, there are several comments need to be addressed.

1. the authors mentioned $> 100 \text{ m}^2$ deposition, which is a key point of this work. However, this has not been demonstrated. All the photos and SEM images are small. Even though it is conceivable that the method COULD be scalable, it is unclear if the performance could still be the same for large scale

deposition (say 1 m^2). this has to be established in this work because the application is a key point of the work.

2. silica nanoparticles will not be very stable for long time at high temperature. Its glass transition temperature is around 1000 deg C. In fact, the ageing data also suggests this points. The stability issue needs to be solved and thoroughly discussed, especially the sintering of the particles (between particles as well as between the particles and the underlying materials) at high T for long period of time, which may make the nanoparticles lose their nanoscale features and thus the light trapping effects. see fig. 4e & 5b

3. In fig. 2, the authors need to discuss why the dummy material has relatively high reflectance, even though the refractive index is like air and it is lossy (absorptive).

4. In the drone experiment, would the concentrated solar heating be sufficient to cure and anneal the NP coating? the color heating will not be as controllable as in the lab.

5. related to the scalability issue, the cost of the silica nanoparticles could be a concern. the authors need to evaluate the economic potential of this method, which I guess is not too great.

Reviewer #3 (Remarks to the Author):

This paper described a very useful CST coating. While the developed coatings are excellent, I felt that they contribute to the development of engineering and not discuss new scientific contributions, which is not in keeping with the philosophy of Nature Communications. The content is excellent and there are probably other magazines that would be more suitable. It is not acceptable for publication, at least in its current state.

1. I cannot understand the scientific novelty of this study. The most important finding of this study is the composition of the matrix. You should show the new scientific discovery.

2. Is there sufficient discussion on particle optimization? You have employed monodisperse spherical silica nanoparticles, but there are many other factors that could be considered, such as other materials, polydisperse, and combinations with other shapes.

3. Although drone-assisted deposition is emphasized in Introduction and Conclusions, it is not discussed. It is only introduced in Methodology. How is this scientifically significant?

Response to Reviewers

We thank all reviewers for their time and effort put into reviewing our manuscript. We greatly appreciate their comments and suggestions and have considered them in the current revision.

The major changes are outlined below:

- We conducted additional modelling to investigate the effect of polydispersity (*versus* monodispersity), non-spherical geometry (*versus* spherical) and material composition (*versus* only silica) – In response to Reviewers 1 and 3.
- We conducted additional field testing to demonstrate that our drone-assisted nanolayer deposition is scalable (i.e. surface areas greater than 1 m²). Supplementary Note 2 and Supplementary Video S1 provide details on the scalability testing – In response to Reviewer 2 and 3.
- We demonstrate experimentally that the nanolayer does not affect the thermal emittance properties of the underlying material. To achieve this, we conducted additional optical characterisation at elevated temperatures. Supplementary Note 3 was added. – In response to Reviewer 1.
- We now provide details on the validation and verification of our numerical modelling. Supplementary Note 1 was added. – Although not in response to a reviewers' comment, this information strengthens confidence in our numerical modelling and provides details on its accuracy.

We look forward to hearing from you and will act swiftly to address further comments if needed.

Response to Reviewer 1

In the submitted manuscript entitled “Scalable Nano-architecture for stable near blackbody solar absorption at higher temperatures”, the authors explored the applicability of the layer nano-architecture on solar absorbers by electromagnetic modelling. Experimentally demonstrated by application on well-established solar absorber coatings like Pyromark & coral-based absorber coatings and enhances the solar absorptance to 35% and stable over 1000h at 900 °C in air. Successfully demonstrate the scalability of the coating application in real field conditions to improve and maintain the solar absorption property of the CST absorber. This manuscript can be accepted after the necessary modifications with answers to all the review comments. Here I am listing a few remarks on reviewing this manuscript:

Response:

We thank the reviewer for their positive feedback. The five points raised by the reviewer were carefully considered in the revised manuscript. Please find below our response to each of these points with the corresponding revision.

1. What was the thickness of the nano-particle layer applied on the base solar absorber layer [and how was estimated?]

Response (coating thickness):

Reviewer 1 raises an excellent point not discussed in detail in the original manuscript.

Our nanolayer has an irregular thickness, varying from the smallest thickness that equates to the matrix thickness (approximately 1–3 nm in our experiments) and the largest effective diameter that equates to the nanosphere diameter plus the matrix thickness (approximately 124 nm in the experiments with best performance).

Fig. 1a now indicates that the nanolayer thickness is less than 250 nm, which is the maximum thickness that yields a nanolayer effectiveness greater than 70%, as per numerical results in **Fig. 3a**. An improvement to **Fig. 1a** was made to clarify the nanolayer thickness.

Revision:

- In line 39, we indicated the thickness of the nanolayer:

the nanolayer is ultrathin, having a thickness normally smaller than 250 nm, i.e. less than 2.5% of the absorber material thickness.

- “< 250 nm” was added to **Fig. 1a** (as shown below) after removing “50–200 nm”.

Response (thickness measurement method):

The coating thickness was measured from SEM images.

As discussed in the original submission, the SEM images show that the matrix with polymer TMOS (**Fig. 5b.1, 2**) is thicker than the matrix with monomer TEOS (**Fig. 4e**). This is a qualitative observation. A quantitative analysis may be done by obtaining the particle size distribution (**Fig. S9d, e**).

As shown in **Fig. S9b** (improved in this revision), the effective nanosphere diameter is larger than the actual nanosphere diameter (pre-mixing in nanolayer solution). In the revised version, we report a quantitative analysis on the effective diameter of the nanospheres with both matrices (thick and thin), as shown in **Fig. S9e**. We now confirm our qualitative observation that the matrix for the polymer TMOS is thicker than that of monomer TEOS. However, the quantitative measurement of the matrix thickness may be inaccurate, so was left out of the main manuscript. This method assumes that the matrix is uniformly covering the nanosphere, which is unlikely to occur due to surface tension effects. The matrix thickness could be estimated as

$$H = (D_e - D)/2,$$

where H is the matrix thickness, D_e is the effective nanosphere diameter, and D is the actual nanosphere diameter (pre-mixing). The thickness of the thin and thick matrices was very similar, within only 3 nm (based on the above equation). However, this approach hypothesises that the matrix covers evenly all the nanosphere, which is hard to confirm given that the cross-section SEM/EDS does not have enough resolution (as previously shown, Torres *et al.*, 2022) for visualising the matrix morphology. We conducted cross-section SEM analyses in an attempt to determine the thickness of the nanolayer including the matrix, but the contrast between matrix and underlying material was not clear. Furthermore, energy-dispersive X-ray spectroscopy (EDS) could not be used for Si elemental mapping due to the rather large interaction volume of the X-ray.

Revision:

- In line 482 (Methods section):

The nominal diameter of silica nanospheres in the colloidal mixture is specified by the manufacturer: IPA-ST-ZL (nominal nanosphere diameter of 80 nm), IPA-ST-L (45 nm nominal) and IPA-ST (12 nm nominal). The measured nanosphere diameter before mixing in the nanolayer mixture and after deposition differs (Fig. S9). Although a direct thickness measurement of thin matrices is challenging, e.g. via cross-section SEM, an indirect measurement can be devised assuming a simplified nanolayer morphology (Fig. S9b). Here, we subtracted the effective nanosphere diameter pre-mixing (without matrix) from the post-deposition effective diameter (Fig. S9d). A uniform matrix thickness around the real nanosphere is assumed to estimate the matrix thickness. This method confirms that the TMOS-based matrix is thicker than the TEOS-based matrix. In addition, a different number density of silica nanospheres can be obtained by increasing or decreasing the number of spray depositions. The coverage after deposition was calculated using SEM images and Eq. (5), where D is the average measured diameter of the nanospheres.

- In the Supplementary Information, we revised the schematic diagram in Fig. S9b, now showing details of the cross-section morphology of the nanolayer used to estimate the matrix thickness.
- We also revised the size distribution histogram and fitted Gaussian curves (normal distribution) in Fig. S9e, showing the comparison of nanolayers with the thick matrix and the thin matrix.

2. Since the developed material applies to CST applications, why was only the solar absorptance property analyzed alone? What about the emittance of the coatings?

Response:

We thank the reviewer for this valuable comment. Emittance was not included in our original manuscript submission due to the following reasons:

1. In moderate-temperature CST applications (e.g. parabolic trough), the emittance is important when introducing spectral selectivity or developing a selective coating, i.e. increasing solar absorption in the visible range while reducing thermal emission in the infrared range. However, it has been reported that for high-flux high-temperature applications, highly absorptive coatings can achieve the same or even better performance than spectral selective ones. (López-Herraiz et al., *Sol. Energy Mater. Sol. Cells*, **159**, 2017 & Noč et al., *Sol. Energy Mater. Sol. Cells*, **238**, 2022). In Fig.6 and relevant discussion of López-Herraiz et al., it is shown that the absorptivity (absorptance) has a much higher impact on the thermal efficiency than emissivity (emittance) in a high-flux high-temperature CST receiver. Therefore, in this work, we focus on increasing the sunlight absorptance of the coating (before and after ageing) for the entire wavelength spectrum to maximise photo-thermal energy conversion, even if there is a penalty due to thermal emission. This is now demonstrated in our Supplementary Note 3 (newly added)
2. In other studies, e.g. Rubin et al., *Sol. Energy Mater. Sol. Cells*, **195** (2019), the nanoporosity extends throughout the entire coating. In contrast, the proposed layer nano-architecture—or “nanolayer”—is an additional nano-texture that is expected to have a marginal effect on the solar absorption spectrum in the infrared range because it is ultrathin (and potentially transparent against infrared light). This nanolayer is intended to be deposited on existing solar absorber coatings, such as Pyromark, to improve their solar absorption (and emittance based on Kirchhoff's law) mostly in the visible range. Control over the infrared spectral absorptance/emittance would require structures with greater length-scales beyond the size of our nanolayer.

However, thanks to your comment, we introduced a discussion on the emittance in the revised manuscript. We now report new experimental results (Fig. S17 and Supplementary Note 3) demonstrating that the nanolayer has some effect on thermal emittance but minor effect on photo-thermal energy conversion. We used a custom-made high-temperature FTIR (Fourier transform infrared spectroscopy) system to measure the emissive power of the absorber with and without nanolayer at elevated temperatures (up to 600°C). The results confirm that our developed nanolayer has an improvement in solar absorptance accompanied by an increase in photo-thermal energy conversion of the same magnitude (Table S1).

Revision:

- In line 376 (paragraph just before the Conclusions section):

It is worth noting that a nanolayer-based improvement in solar absorptance (from a baseline solar absorptance of ca. 97%) is expected to increase the thermal emittance due to Kirchhoff's law, which states an equivalence of spectral emittance and spectral absorptance at thermal equilibrium conditions. An increase in thermal emittance on high-temperature surfaces results in a greater radiative heat loss due to the Stefan–Boltzmann law. However, most central tower CST applications have sunlight concentrations exceeding 1000 suns, meaning that the negative impact on receiver efficiency from the high emittance is much smaller than the positive effect from a high solar absorptance [47]. In addition, nanoscale features in our nanolayer (ca. 120 nm) are much smaller than the dominant wavelength of the thermal emission (ca. 3000 nm for a blackbody at

700 °C, based on the operating temperature of CST receivers). Therefore, nanoscale features cannot be tuned to effectively control thermal emission in the infrared range. We measured the relative increment of emittance at high temperatures after nanolayer deposition (Supplementary Note 3) and confirmed an increase of ca. 1.4%. However, its impact on photo-thermal energy conversion is negligible (as assessed by a *figure of merit*, Table S1) when deposited on a solar absorber coating exposed to sunlight concentration of 1000 suns.

- We added Supplementary Note 3 with a discussion on the impact of our nanolayer on the thermal emittance (measured at high temperatures) when deposited on a solar absorber coating. Below we include the results figure and table for your reference.

Fig. S17 | caption in *Supplementary Information*.

Table S1 | caption in *Supplementary Information*.

Conditions	α (%)	$\alpha_{nl} - \alpha_{sm}$ (%)	ϵ (%)	$\epsilon_{nl} - \epsilon_{sm}$ (%)	η (%)	$\Delta\eta = \eta - \eta_{sm}$ (%)
Solar material (sm) without nanolayer	96.82	–	88.18	–	92.24	–
Solar material with nanolayer (nl)	97.36	+0.54	89.52	+1.34	92.72	+0.47
increasing α ; unchanged ϵ	97.36	+0.54	88.18	–	92.78	+0.54
unchanged α ; increasing ϵ	96.82	–	89.52	+1.34	92.18	–0.07

3. How to do the coverage & particle size of the nano-spheres on the base absorber materials affect the enhancement of solar absorptance? How is 30% coverage optimized?

Response:

Thank you for these questions. The related results and discussion can be found in Section 2.4, second paragraph (*the effect of particle size and coverage*) and third paragraph (*how was the optimal 30% coverage obtained*), both presented in relation to Fig. 4. We amended the manuscript to some extent such that these results are presented more clearly.

Revision:

- In line 259 (second paragraph of Sec. 2.4):

These experimental results show a good qualitative agreement with our theoretical predictions (Fig. 2a), i.e. the absorptance increases with nanosphere diameter from 12 nm until a near-optimal improvement with nanospheres of ca. 117 nm in diameter (Fig. S9e shows the measured size distribution). Likewise, the experimental results in Fig. 4b show that an increased coverage from 10.6 % to 60.7 % yields larger solar absorptance, which is again in agreement with our theoretical predictions (Fig. 2d).

4. Does the wide distribution of Nano-spheres (11 – 117 nm) influence the enhancement of the solar absorptance in the experimental section, whereas, in modelling, nanosphere size is considered?

Response:

This is an excellent point raised by the reviewer, which was not discussed in the original manuscript. It is worth clarifying that in our experiments, there is no “wide distribution” of nanosphere particle spanning from 11 nm to 117 nm, i.e. the polydispersity in our experimental nanolayer is rather moderate. This is now shown quantitatively by a Gaussian fit of the nanosphere size distribution in Fig. S9e. Nonetheless, polydispersity does exist and, therefore, in the revised manuscript we now report modelling results for the same random configuration as in original Fig. 2a but with polydisperse nanospheres. The new results are now included in Fig. 2b (the previous Fig. 2b was merged into Fig. 2a) and the newly added Supplementary Fig. S2. We also added a discussion on the effect of polydispersity on coverage.

Revision:

- In line 129 (second paragraph of Sec. 2.2):

Polydispersity in the nanosphere arrangement has generally a marginal effect on the nanolayer effectiveness, as shown in Fig. 2b. A mild downward trend of effectiveness is observed when increasing the standard deviation from the monodisperse condition to a relative standard deviation ($\sigma/D_{\text{ave}} \times 100\%$, where σ is the standard deviation of the particle size distribution and D_{ave} is the mean nanosphere diameter). For the mean diameter close to the optimal size of 100 nm, the absorptance did not change significantly with polydispersity within a relative standard deviation of 10%. A condition close to a monodisperse arrangement seems beneficial, which is confirmed for the dummy material when introduced the underlying absorber (Fig. S2b,c shows results for other solar absorbers). This slightly downward trend is due to a reduced resonance effect (lower absorption peak) for polydisperse nanospheres compared with a monodisperse distribution (Fig. S2a shows spectral absorptance). In some cases, a slightly higher effectiveness is observed for nanospheres with mild polydispersity (< 5% in relative standard deviation) but the change is marginal, i.e. $e_{\text{nl}} < 1\%$. Therefore, the simulations that follow in this study consider monodisperse nanospheres.

- In line 184 (last paragraph of Sec. 2.2):

It is worth noting that, for the same number of nanospheres, the polydisperse distribution would statistically introduce a higher coverage than the corresponding monodisperse case with the same mean particle size. However, we found that an increased degree of polydispersity (measured by the relative standard deviation) does not affect the wavelength at which there is a peak in spectral absorptance (Fig.S2a), yet the absorption peak slightly drops.

- In line 449 (second paragraph of Sec. 4.2):

A normal (Gaussian) distribution was used for the simulations with polydisperse nanospheres. Although a lognormal distribution is generally more suitable than a normal distribution for describing the polydisperse nanoparticles, our particle size characterisation (Fig. S9d,e) indicates that nanolayers with nanospheres of mean diameter above 100 nm do follow a normal distribution. We note that distributions with particles smaller than 50 nm in mean diameter do follow a lognormal distribution. In the modelling, to avoid potential negative diameters in a normal distribution, the minimum diameter was set to zero. A representative distribution of polydisperse nanospheres is shown in Fig. S2a.

- New results for polydisperse distribution were added to **Fig. 2b** (caption in *manuscript* file):

- New results for polydisperse distribution in **Fig. S2** (caption in *Supplementary Information*):

5. Provide legends in Fig S4 (b).

Response:

We thank the reviewer for this suggestion. **Fig. S6b** (previously Fig. S4b) now labels each curve and indicates the corresponding nanosphere layout (randomly generated) *in lieu* of using a legend.

Revision:

- We modified **Fig. S6b**:

Response to Reviewer 2

The authors showed a scalable method using silica nanoparticle deposition (nanolayer) to enhance the solar absorptance of state of the art solar absorbing materials such as Pyromark. The work is generally interesting. While the principles reported here are not quite new (using light trapping effect of nanoparticles), the application of using this method for potentially large scale deposition on solar absorbing materials is novel. However, there are several comments need to be addressed.

Response:

We thank the reviewer for the positive feedback. The five points raised by the reviewer were carefully considered in the revised manuscript. Please find below our response to each of these points with corresponding revision.

1. The authors mentioned $> 100 \text{ m}^2$ deposition, which is a key point of this work. However, this has not been demonstrated. All the photos and SEM images are small. Even though it is conceivable that the method COULD be scalable, it is unclear if the performance could still be the same for large scale deposition (say 1 m^2). This has to be established in this work because the application is a key point of the work.

Response:

We thank the reviewer for this valuable comment. We acknowledge that the photo previously shown in **Fig. 5d.2** (original submission) was insufficient to demonstrate scalability. To address this point, we conducted additional scalability experiments and now report results that demonstrate, with photos and videos, that the drone-assisted deposition experiment yields a scalable deposition of our proposed nanolayer (exceeding 1 m^2). Two sets of hotplates with flat and curved samples were prepared to determine if the evaporated solvent from the deposition would influence the adjacent surfaces. We confirmed that the spectral absorptance and surface morphology were the same as those samples deposited in a more controlled lab environment. Furthermore, since the reproducibility of the drone experiment was excellent with all samples exhibiting very close absorptance, only representative results are shown in **Fig. 5b, e**. More detailed information of the drone deposition experiment is now given in the newly added Supplementary Note 2 and Video S1.

Revision:

- In line 374 (Sec. 2.4):
(see Supplementary Note 2 for more details on the nanolayer scalability experiments using a drone)
- We added Supplementary Note 2 with a technical note on the scalability testing and Supplementary Video S1 showing different drone-assisted nanolayer deposition experiments. Please see the *Supplementary Information* for details. Below we copy-paste the figure for your reference.

Fig. S17 | caption in *Supplementary Information*.

2. Silica nanoparticles will not be very stable for long time at high temperature. Its glass transition temperature is around 1000 deg C. In fact, the ageing data also suggests this point. The stability issue needs to be solved and thoroughly discussed, especially the sintering of the particles (between particles as well as between the particles and the underlying materials) at high T for long period of time, which may make the nanoparticles lose their nanoscale features and thus the light trapping effects. see fig. 4e & 5b.

Response:

Thank you for this important comment. A more in-depth discussion on the stability of the silica nanoparticles was missing in the original submission. We now emphasise on (1) the stabilisation strategy and (2) materials selection. We also note that (3) the results presented in the original submission emphasised the stability of the nanolayer even after extensive annealing.

1. **Stabilisation strategy.** As pointed out by the reviewer, sintering of particles is a big issue for a reliable implementation of the proposed nanolayer material at high temperatures for long exposure time. To mitigate this problem, prior to deposition, the nanoparticles are well dispersed within an organic solution containing a silica precursor (either unpolymerised or polymerised precursor) to form the nanolayer mixture. This meticulous preparation prevents the agglomeration of nanospheres within the mixture. Therefore, when the nanolayer mixture is deposited, a silica matrix that binds the nanoparticles to the substrate is formed with the well dispersed nanospheres (e.g. large clusters of nanospheres are not observed in **Fig. 4e**). However, we acknowledge that

complete avoidance of nanosphere contact, particularly at higher coverage, is challenging to achieve.

2. **Material selection.** The solar absorber materials that our nanolayer aims to enhance are all operated well below the glass transition temperature of pure silica (approximately 1200 °C). For example, the surface temperature range of a conventional receiver in a central tower concentrating solar power (CSP) plant generally falls between 600 °C and 700 °C. In fact, we use a higher temperature (900 °C) in our experiments to accelerate the kinetics and shorten the ageing time.

In the original submission we stated that silica is a highly stable ceramic:

In line 50: “In this work, we apply a controllable and scalable layer nano-architecture made of silica (i.e. a highly-stable ceramic at elevated temperatures) onto several arbitrary high-temperature solar absorber materials...”

3. **Results.** In the original submission, we commented on the SEM images, which showed the morphology of the nanospheres after extensive (1000 h) thermal annealing at elevated temperatures (900 °C). There is no obvious sintering from the SEM analysis for our nanolayer material, in contrast to what has been observed for other types of solar absorber materials (Torres et al. *Sol. Energy Mater. Sol. Cells*, **218** 110719, 2020).

In line 293: “Minor morphology changes are observed in the nanospheres after ageing at 900 °C for 100 h, 400 h (Fig. S10) and 1000 h (Fig. 4e), regardless of the coverage.”

We would like to clarify that the SEM image in Fig. 5b.2 shows the nanolayer with thick matrix (polymer TMOS) after ageing for 1000 h at 900 °C. This is not sintering of the nanoparticles with the underlying material but an immersion of the nanoparticles into the matrix. The pristine condition is shown in Fig. 5b.1.

Revision:

- In line 297 (Sec. 2.4), regarding the strategy for nanolayer **stabilisation** (i.e. avoiding sintering):

It is worth noting that prior to deposition, the nanoparticles are well dispersed within an organic solution containing a silica precursor that forms the matrix after the nanolayer mixture is deposited (see Methods section for details). This mixture preparation prevents the agglomeration of nanospheres and thus, after deposition, it lowers the likelihood of nanosphere sintering when exposed to high temperatures for a long time [27].

- In line 301 (Sec. 2.4), regarding the **materials selection** for nanolayer (i.e. avoiding phase transition):

Furthermore, the solar absorber materials that our nanolayer aims to enhance are all operated well below the glass transition temperature of pure silica (ca. 1200 °C). In contrast, the surface temperature range of a conventional CST receiver generally falls between 600 °C and 700 °C. In fact, we use a higher temperature (900 °C) in our experiments to accelerate the kinetics and shorten the ageing time.

3. In Fig. 2, the authors need to discuss why the dummy material has relatively high reflectance, even though the refractive index is like air and it is lossy (absorptive).

Response:

We thank the reviewer for this suggestion. The dummy material has relatively high reflectance because the dummy material does not have any roughness, at any length scale. The reflection coefficient at the interface between the material and air can be calculated *via* Fresnel equations and thus the “reflectivity” (not “reflectance”) is the square of the reflection coefficient (these equations can also be found in the newly-added Supplementary Note 1).

The statement in the original manuscript that the dummy material has “a refractive index close to that of air” was imprecise. In fact, the refractive index of the dummy material is between 10% ($n = 1.1$) and 80% ($n = 1.8$) greater than that of air, as shown in Fig. S1a (figure below), which could still be considered high relative to air. However, tungsten is known to be a good absorber, yet its refractive index is higher than our dummy material ($n > 3$ in the visible range) as shown in Fig. S1c. The revised manuscript now clarifies that the dummy material has a refractive index “relatively close to that of air”, but still with a tangible reflection as shown in Fig. S1b.

Revision:

- In line 120 (first paragraph of Sec. 2.2):

Here, the solar absorber is mimicked by a dummy material with a refractive index relatively close to that of air (antireflective) and a moderate extinction coefficient (Fig. S1a). These properties were chosen to produce a spectral absorptance close to that of Pyromark (Fig. S1b) despite not having any roughness.

- We added the spectral absorptance of the dummy material and recently optimised coating of Pyromark 2500 as the reference in Fig. S1b:

b. Spectral absorptance of the dummy material. The spectral absorptance of recently optimised coating of Pyromark 2500 [1] is included here as the reference. Note that the reflectance of the dummy material is still relatively high because it is a theoretical result based on the Fresnel equations for a perfectly flat surface (Supplementary Note 1), whereas the results for Pyromark are experimental and for a highly porous composite.

4. In the drone experiment, would the concentrated solar heating be sufficient to cure and anneal the NP coating [nanolayer]? the color (*solar) heating will not be as controllable as in the lab.

Response:

Thank you for this question. The curing and thermal annealing of the nanolayer do not use concentrated solar heating as implied in the reviewer's question. In a practical application the tube banks of a solar thermal receiver can be kept at the temperature required (up to 350°C) by blending and circulating molten salt from hot and cold salt tanks. In the drone experiment, the hotplate is under the samples to mimic a similar temperature as the receiver surface. Details on the drone experiment are now included in the newly-added Supplementary Note 2 (accompanied by Video S1). The nanolayer mixture can be rapidly sprayed onto heated receiver surfaces using a drone (when not irradiated with concentrated sunlight).

Revision:

- We included this information in Supplementary Note 2. Please see the *Supplementary Information* for details.

Supplementary Note 2: Scalability experiment for drone-assisted nanolayer deposition

5. Related to the scalability issue, the cost of the silica nanoparticles could be a concern. the authors need to evaluate the economic potential of this method, which I guess is not too great.

Response:

We thank the reviewer for this valuable comment.

We have not mentioned costs in the manuscript due to non-disclosure agreements with industry partners. However, we can state that our nanolayer is less expensive than the underlying solar absorber materials that we explored, paving the way towards the commercial viability and implementation of the nanolayer. Take the example of repainting Pyromark on degraded Pyromark coating: the material cost of Pyromark is $\sim \$5.41/\text{m}^2$ (C.K. Ho and J.E. Pacheco, *SANDIA REPORT*, 2013). We found that the cost per square meter for materials of our nanolayer is less than those for Pyromark 2500. The reason for this economic outlook is the low cost of raw materials in our nanolayer mixture. Most of its composition is industrial ethanol (the detail can be found in the Sec. 4.3) and the silica nanospheres are already being mass produced at low costs (e.g. all our materials come from standard industrial processes). Furthermore, coating deposition is also more inexpensive for the nanolayer than for Pyromark 2500 despite costs associated with heating the underlying surface. A significant advantage of the nanolayer is its low thermal resistance, allowing us to coat the nanolayer on the solar absorber without the need to physically remove the degraded coating (e.g. scrapping or sandblasting), which is generally very expensive. Note that if Pyromark is coated on degraded Pyromark without physical removal, then the thermal resistance could be excessively large because the coating thickness could be excessive, leading to prohibitive radiative heat losses (for a given heat transfer fluid temperature – see discussion in the Supplementary Note 7 in the Supplementary Information of Torres et al. *Energy Environ. Sci.*, 2022, **15**, 1893).

Revision:

- In line 505 (second paragraph of Sec. 4.3.2):

All materials used to produce the nanolayer mixture come from proven industrial processes that are suitable for mass production. This includes the silica nanospheres, industrial ethanol and silica precursors (TEOS and TMOS). Furthermore, the nanolayer has much less mass than that of the underlying solar absorber coating (**Fig. 1a**), so material costs are reduced. Furthermore, we have shown theoretically that our layer nano-architecture is not very sensitive to its dimensions (**Fig. 3a**) such as nanosphere size ($e_{\text{nl}} > 70\%$, 100–150 nm), matrix thickness ($e_{\text{nl}} > 70\%$, 30–90 nm), immersion ($e_{\text{nl}} > 70\%$ for $\zeta < 20\%$) and polydispersity (**Fig. 2b**). The lax requirement for precision in manufacturing is another contributing factor to lowering costs.

Response to Reviewer 3

This paper described a very useful CST coating. While the developed coatings are excellent, I felt that they contribute to the development of engineering and not discuss new scientific contributions, which is not in keeping with the philosophy of Nature Communications. The content is excellent and there are probably other magazines that would be more suitable. It is not acceptable for publication, at least in its current state.

We thank Reviewer 3 for their time reviewing our manuscript. Based on their valuable comments, the manuscript was carefully revised. Noting the positive feedback from Reviewers 1 and 2, we believe there is strong evidence that our paper is of the high standards required by *Nature Communication*.

Furthermore, the Editor has kindly advised that applied science and engineering research is also of interest to *Nature Communications* and publication does not require a new scientific discovery. This journal policy is quoted below from the following source: <https://www.nature.com/ncomms/submit/applied-science-research>

“*Nature Communications* acknowledges that when assessing application-oriented manuscripts, the level of novelty will likely be below that typically observed in more fundamental work. Instead, the advance comes from the performance or utility of the research. These cases require shifting the focus of evaluation to technology readiness and its potential to overcome existing real-world challenges, even when all the fundamentals involved are well reported in the academic literature.”

The Editor has confirmed that the journal does not require revision action for the first and third points, but the second point must be considered. We kindly request Reviewer 3 to reassess their recommendation based on the response below and the revised manuscript.

1. I cannot understand the scientific novelty of this study. The most important finding of this study is the composition of the matrix. You should show the new scientific discovery.

Response:

We respectfully disagree with this comment. The matrix composition fits within one of several major contributions of our paper. Three major contributions are outlined below.

1. **Conceptually**, we propose and develop the first layer nano-architecture—or “nanolayer”— for enhanced solar absorption in high-temperature, high-flux environments. This will enable advanced concentrating solar technologies (CST), improving performance and facilitating their maintenance. Conceptually, this has never been proposed or implemented before.
2. **Scientifically**, our paper advances understanding on the following:
 - a. The effect of random *versus* uniform dispersion of nanospheres, the former being representative of our nanolayer.
 - b. The effect of nanoparticle size and density (coverage). Importantly, this is tackled experimentally and numerically.
 - c. Interplay between nanosphere size, matrix thickness, and particle immersion.
 - d. Materials and optical stability for the silica nanolayer when deposited on two distinct solar absorber materials.
 - e. Effect of polydispersity *versus* monodisperse nanospheres (newly added content), the former being representative of our nanolayer. This new understanding is supported by theoretical results based on solving Maxwell’s equations via computational electromagnetics (CEM).
 - f. Effect of varying nanoparticle shape and materials optical properties.

Based on your second comment, we have added understanding on the last two points (e, f).

3. **Technologically**, our paper showcases the following:

- a. Scalability using drone technology.
- b. Tunability of nanolayer morphology with varying coverage, nanosphere size and matrix thickness.

We believe that the coating development from a theoretical framework to lab-based testing to implementation merits the attention of the broader readership of *Nature Communication*.

As a reference, please find below some recent examples of papers published in *Nature Communications* where a significant contribution to the advancement of engineering is made:

- Xu, G., Overvig, A., Kasahara, Y. *et al.* Arbitrary aperture synthesis with nonlocal leaky-wave metasurface antennas. *Nat Commun* **14**, 4380 (2023)
<https://doi.org/10.1038/s41467-023-39818-2>
- Lu, H., Cui, H., Lu, G. *et al.* 3D Printing and processing of miniaturized transducers with near-pristine piezoelectric ceramics for localized cavitation. *Nat Commun* **14**, 2418 (2023)
<https://doi.org/10.1038/s41467-023-37335-w>
- Ouyang, W., Xu, X., Lu, W. *et al.* Ultrafast 3D nanofabrication via digital holography. *Nat Commun* **14**, 1716 (2023)
<https://doi.org/10.1038/s41467-023-37163-y>
- Dong, W.J., Xiao, Y., Yang, K.R. *et al.* Pt nanoclusters on GaN nanowires for solar-assisted seawater hydrogen evolution. *Nat Commun* **14**, 179 (2023)
<https://doi.org/10.1038/s41467-023-35782-z>
- Sun, Z., Han, C., Gao, S. *et al.* Achieving efficient power generation by designing bioinspired and multi-layered interfacial evaporator. *Nat Commun* **13**, 5077 (2022)
<https://doi.org/10.1038/s41467-022-32820-0>

2. Is there sufficient discussion on particle optimization? You have employed monodisperse spherical silica nanoparticles, but there are many other factors that could be considered, such as other materials, polydisperse, and combinations with other shapes.

Response:

This is an excellent point raised by the reviewer, which was not discussed in the original manuscript. We conducted further simulations to investigate the three factors raised by the reviewer. We conducted a sensitivity test for independent parameters without considering non-linearities that could arise due to a combination of multiple parameter variables.

A. On other materials

Silica is a ceramic, non-toxic material with low refractive index, which can be used for antireflection. In addition, the production of silica nanoparticles in colloidal and powder forms has been demonstrated at an industrial scale. Silica nanoparticles can be synthesized by various methods including ball milling, hydrothermal, chemical vapor deposition, microemulsion, and sol–gel. For the purpose of obtaining silica particles with uniform size, high purity, ease to control, and scalability, the sol–gel process is the most popular. Sol–gel is a synthesis method that is suitable for industrial manufacturing due to its simplicity, homogeneity, and refined result. The importance of the sol–gel process in the formation of the silica particle structure has been studied by Zulfiqar *et al.* “Synthesis of silica nanoparticles from sodium silicate under alkaline conditions.” *J Sol-Gel Sci Technology*. **77**, 753–758 (2016).

Therefore, in the original submission, we only considered silica as the nanolayer material (nanosphere and matrix) in both simulation and experiments. However, as pointed out by the reviewer, it is worth considering other materials for both theoretical insight and potential improvement to nanolayer effectiveness. Hence, we conducted additional modelling with the same random configuration as in **Fig. 3a**, but now using with alumina (Al₂O₃) and titania (TiO₂) as nanolayer materials. These materials

have been used for nanostructures in the photovoltaic industry. Since these two materials are not used in our nanolayer in the experiments, we added these results in the *Supplementary Information Fig. S6* and an associated discussion in the manuscript. The theoretical prediction of reflectivity (matrix only) or scattering efficiency (sphere only) for different materials with distinct optical properties can be calculated with the equations showing in *Supplementary Note 1*, a newly added technical note for the validation and verification of our FDTD simulations.

Revision:

- In line 213, (first paragraph of Sec. 2.3):

A further extended simulation regarding different materials of nanolayer (both nanosphere and matrix) is conducted. Although materials with a higher refractive index (e.g. $\alpha\text{-Al}_2\text{O}_3$) may introduce more reflectivity when used as a layer (matrix) without nanospheres, a larger light scattering by the nanospheres may occur (based on Mie theory). Thus, a slightly greater effectiveness is observed for alumina (*Fig. S8*) than silica (*Fig. 3a*) when deposited on the dummy material. However, a material with a larger refractive index than that of the underlying material (e.g. rutile TiO_2 vs. dummy material) may increase light reflection ($e_{\text{nl}} < 0$) due to the occurrence of total internal reflection for scattered light with an angle of incidence onto the underlying material greater than a critical value. Silica seems to be a suitable choice for the nanomaterial because of both its broad range improvement in effectiveness (always $e_{\text{nl}} > 0$ for the studied parameter range; *Fig. 3a*, *Fig. S7*) and its industrial scale production.

- We added these results in *Fig. S8*

Fig. S8 | caption in *Supplementary Information*.

B. On polydisperse nanospheres

This is an excellent point raised by the reviewer, which was not discussed in the original manuscript. Polydispersity does exist as shown in our experimental nanolayer (*Fig. S9d,e*) and, therefore, in the revised manuscript we now report modelling results for the same random configuration as in original *Fig. 2a* but with polydisperse nanospheres. The new results are now included in *Fig. 2b* (the previous *Fig. 2b* was merged into *Fig. 2a*) and the newly added *Supplementary Fig. S2*. We also added a discussion on the effect of polydispersity on coverage.

Revision:

- Same modifications as those in response to Reviewer 1 Comment 4 (pp.6–7 of this response file).

C. On the combinations with other shapes

Nanoparticles can have many shapes. As mentioned in the introduction, the advantage of utilising spherical nanoparticles, as opposed to other shapes, lies in that the sphere diameter is a parameter easily controllable. Therefore, the scope of our simulations is tailored to the geometry of our proposed layer nano-architecture with spherical nanoparticles (plus matrix) and excluding shapes with edges like prisms, rectangles or cylinders. Nonetheless, based on the reviewer's comment, we explored the inclusion of nanoparticles with an ellipsoidal shape as a representative irregularity in a near-spherical nanoparticle.

The simulation was set with a uniform configuration on the dummy material. The results suggested that elongating one of the ellipse axes will improve the solar absorptance, with the spectral absorption peak increasing and shifting to the infrared range. It is worth noting that the change of shape can significantly affect the coverage. These outcomes, observed through adjustments in ellipse shapes, are very similar to the coverage effect results of **Fig. 2d, e**.

Revision:

- In line 188 (last paragraph of Sec. 2.2):

In addition, the potential irregular shape of nanoparticles is also considered by assuming an ellipsoidal particle shape. Interestingly, the results of the effectiveness and spectral absorptance for monodisperse ellipsoids (with uniform configuration, **Fig. S5**) are similar to the monodisperse nanospheres with variable coverage shown in **Fig. 2d,e**, suggesting a correlation between ellipsoid parameters and coverage.

- New results for ellipsoidal particle shape in **Fig. S5** (caption in *Supplementary Information*):

3. Although drone-assisted deposition is emphasized in Introduction and Conclusions, it is not discussed. It is only introduced in Methodology. How is this scientifically significant?

Response:

We thank reviewer for this comment.

In the original submission, the main text briefly discussed the drone-assisted deposition:

In line 369: “Furthermore, **Fig. 5d** shows the concept of spraying the improved nanolayer (with a polymerised silica matrix) using a UAV and the scalability testing on heated samples. The resulting improvement in the pristine condition is shown in **Fig. 5e**,...”

However, we acknowledge that the photo previously shown in Fig. 5d.2 (original submission) was insufficient to demonstrate the engineering significance. Since the drone deposition itself is a method, we have not provided more information in the main manuscript. Nonetheless, based on this comment and Reviewer 2 Comment 1, we added additional information in the *Supplementary Information* in the form of Supplementary Note 2 and Video S1.

We conducted additional experiments and now report results, with photos and videos, demonstrate that the drone-assisted deposition experiment yields a scalable deposition of our proposed nanolayer. Two sets of hotplates with flat and curved samples were prepared to determine if the evaporated solvent from the deposition would influence the adjacent surfaces. We confirmed that the spectral absorbance and surface morphology were nearly identical to those samples deposited in the lab. Furthermore, since the reproducibility of the drone experiment was excellent with all samples exhibiting very close absorbance, only representative results are shown in **Fig. 5b,e**.

Regarding the “scientifically significant” aspect, recall the *Nature Communications* guidelines on applied science and engineering research (in the response to your Comment 1). The journal considers articles reporting research that narrows the gap between academic knowledge and real-life applications, from proof-of-concept studies for innovative solutions and designs to large-scale technical demonstrations of capability. Our drone experiments fit this description.

Revision:

- The photo in **Fig. 5d.2** was changed to better show the drone experiment.

- We added Supplementary Note 2 with a technical note on the scalability testing and Supplementary Video S1 showing different drone-assisted nanolayer deposition experiments. The figure of Supplementary Note 2 is included in the response to Reviewer 2 Comment 1 as a reference (p.10 of this response file).

REVIEWERS' COMMENTS

Reviewer #1 (Remarks to the Author):

Thank you for the author's answer to the queries raised in the 1st review of the manuscript.

However, the authors did an excellent job comparing the experimental research with the simulation, but still, I am not convinced of scientific novelty, and I fully agree with the 3rd reviewer's statement that the important finding of this study is the composition of the matrix for the enhancement of solar absorptance over on the solar absorber layers. There are numerous studies on the silica and SiO₂ nanoparticles-based optical enhancing layer on the selective absorber layers (Ref.: Sol. Energy 115 (2015) 341–346, Surf. Eng. 32 (2016) 840–845, ACS Appl. Nano Mater. 3 (2020) 7869–7878). In this contest, this work is not suitable for publishing nature communications. The content of this manuscript is more suitable for other magazines!.

Reviewer #2 (Remarks to the Author):

The authors revisions are satisfactory to me. I appreciate the authors' efforts of demonstrating ~1 meter square coating, which I think is important and practically useful.

Reviewer #3 (Remarks to the Author):

The paper has been carefully revised and I am accepting this paper.